# Long-read single-cell sequencing reveals expressions of hypermutation clusters of isoforms in human liver cancer cells

Silvia Liu[1,2,3]*, Yan-Ping Yu[1,2,3], Bao-Guo Ren[1,2,3], Tuval Ben-Yehezkel[4], Caroline Obert[4], Mat Smith[4], Wenjia Wang[5], Alina Ostrowska[1,3], Alejandro Soto-Gutierrez[1,3], Jian-Hua Luo[1,2,3]*

[1]Department of Pathology, University of Pittsburgh, Pittsburgh, United States; [2]High Throughput Genome Center, University of Pittsburgh, Pittsburgh, United States; [3]Pittsburgh Liver Research Center, University of Pittsburgh, Pittsburgh, United States; [4]Element Biosciences Inc, San Diego, United States; [5]Biostatistics, University of Pittsburgh, Pittsburgh, United States

*For correspondence:
shl96@pitt.edu (SL);
luoj@upmc.edu (J-HL)

**Abstract** The protein diversity of mammalian cells is determined by arrays of isoforms from genes. Genetic mutation is essential in species evolution and cancer development. Accurate long-read transcriptome sequencing at single-cell level is required to decipher the spectrum of protein expressions in mammalian organisms. In this report, we developed a synthetic long-read single-cell sequencing technology based on LOOPSeq technique. We applied this technology to analyze 447 transcriptomes of hepatocellular carcinoma (HCC) and benign liver from an individual. Through Uniform Manifold Approximation and Projection analysis, we identified a panel of mutation mRNA isoforms highly specific to HCC cells. The evolution pathways that led to the hyper-mutation clusters in single human leukocyte antigen molecules were identified. Novel fusion transcripts were detected. The combination of gene expressions, fusion gene transcripts, and mutation gene expressions significantly improved the classification of liver cancer cells versus benign hepatocytes. In conclusion, LOOPSeq single-cell technology may hold promise to provide a new level of precision analysis on the mammalian transcriptome.

## eLife assessment

The authors pair single-cell sequencing technology with the LoopSeq synthetic long-read method to examine samples of hepatocellular carcinoma and benign liver, with the goal of identifying mutations and fusion transcripts specific to cancer cells. The authors present a **valuable** resource, and the overall support for the major claims is **solid**.

## Introduction

Mammalian organisms are composed of numerous cells with multiple different roles. Individual cells are supported by a broad array of proteins with a variety of functions. While protein expression is dictated by the level of gene expression, the structure and function of the protein are largely determined by the isoforms of the mRNA of a given gene and are impacted by mutations or other structural alterations to the amino acids (*Faustino and Cooper, 2003*). To understand the role of each cell in an organism, broad-spectrum mRNA isoform and mutational gene expression analyses at the single-cell level are necessary.

Human cancers have been known for their extensive genomic alterations (*Hanahan, 2022*), including mutations, chromosome rearrangements, insertion/deletion, etc. These genome alterations drive the clinical course of the disease through the expression of the mutated transcripts and proteins (*Hollstein et al., 1991*; *Murugan et al., 2019*). Even though the mutations or mutation clusters have been clearly demonstrated in the genome level (*Bergstrom et al., 2022*; *Gerstung et al., 2020*; *Nam et al., 2021*), it was unclear whether these mutations were expressed in the RNA transcripts and in what protein isoforms if they were expressed. Furthermore, it was unclear whether the mutations occurred in the same allele if multiple non-adjacent mutations were detected.

In the last 10 y, great strides in the field of long-read sequencing have enabled the quantification of mRNA isoforms in mammalian samples (*Logsdon et al., 2020*; *Nakano et al., 2017*). These sequencing technologies have been successfully employed to quantify mRNA isoforms from the bulk samples (*Athanasopoulou et al., 2022*). However, little progress has been made in developing a technology to analyze mutated mRNA expressions at the single-cell level. Among the long-read sequencing solutions, LoopSeq synthetic long-read sequencing technology has been shown to produce the lowest error rate (*Liu et al., 2021a*) and thus may be most suited for the mutational isoform expression analyses. In this report, we developed a strategy to integrate Element Biosciences' LoopSeq intramolecular barcoding technique with 10x Genomics' cell barcoding scheme to create a single-cell long-read isoform analysis vehicle. To demonstrate the utility of this methodology, we analyzed the isoforms of over 440 transcriptomes from the cells originating from a hepatocellular carcinoma (HCC) patient. The results showed evolutionary patterns of single-molecule mutational gene expression from benign hepatocytes to liver cancer cells.

## Results
### Single-cell LoopSeq strategy
The strategy of incorporating LoopSeq long-read technology with single-cell sequencing starts with utilizing the output of 10x Genomics' 3' single-cell assay. Approximately 200–300 cells from samples of both benign liver or HCC from a patient were encapsulated and unique molecular barcoded using a Gel Beads-in emulsion (GEM) system. The Gel Beads were dissolved, and any respectively associated cells were lysed prior to reverse transcription, template switching, and transcript extension. The

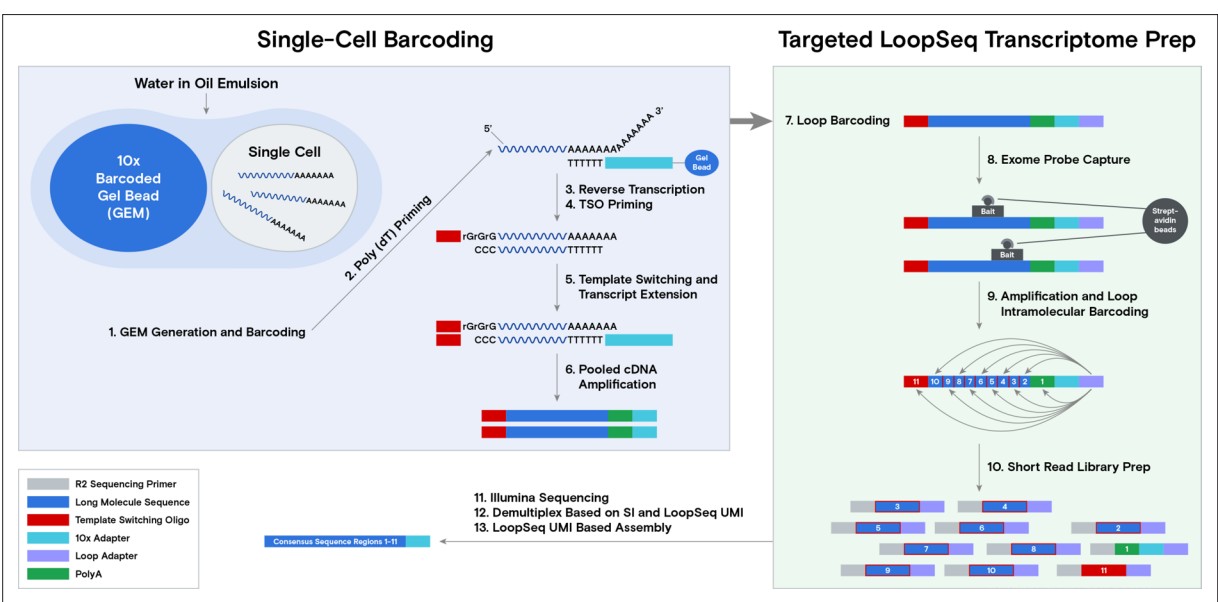

**Figure 1.** Schema of the workflow for the single-cell LoopSeq assay. A total of 200–300 live cells per sample were co-partitioned with Gel Beads and subsequently lysed. The captured mRNAs were reverse-transcribed and barcoded using Chromium Next GEM 3' reagent 3.1 kit (10x Genomics). The cellular barcoded cDNAs were ligated with a LoopSeq Adapter (Element Biosciences) and enriched by human core exome capturing (Twist Biosciences). This was followed by amplification and intramolecular distribution of the LOOP unique molecular identifier (UMI) located on the LoopSeq Adapter. The LOOP UMI barcoded cDNAs were then fragmented and ligated with an adaptor to generate a short-read sequencing library before sequencing.

resulting 10× Adapter sequence contains a 16-base pair barcode, followed by a 12-base pair unique molecular identifier (UMI) and a 30-base pair poly(dT) sequence. Full-length cDNA was then amplified from purified first-strand cDNA (*Figure 1*, blue box). A Loop adapter (containing 12-base pair unique molecular identifiers [LOOP UMI] and a 6-base pair sample index [SI]) was ligated to the 10x Genomics barcoded cDNA and subsequently enriched by exome probe sets that represent 19,433 genes in the human genome. This was followed by amplification and the random intramolecular distribution of the LOOP UMIs throughout their respective cDNA molecules. The LOOP UMI-distributed fragments were then subjected to short-read library preparation for sequencing (*Figure 1*, green box). Binned cDNA short-reads from individual LOOP UMIs were de novo assembled to generate consensus sequences of long mRNA transcripts.

To produce sufficient long-reads for single-cell analysis, 2.985 billion short-reads were sequenced across the benign liver cells, while 3.814 billion short-reads were evaluated from the HCC cells. The assembly of these short-read sequences resulted in 5.8 million long-read transcripts for the benign liver sample and 6 million for the HCC sample. The mapping of 10x Genomics cell barcodes resulted in 447 valid single-cell transcriptomes (162 from the benign liver sample and 285 from the HCC sample). There were an average number of 1186 genes per cell and 1331 isoforms per cell in the benign liver sample and 1266 genes and isoforms in the HCC sample. Interestingly, there was a total of 8646 novel isoforms identified in benign liver tissue (based on SCANTI v1.2). This generated an average of 442 novel isoforms per cell. For the HCC sample, there was a total of 14,229 novel isoforms. This translated into an average of 450 novel isoforms per cell.

## Mutational gene and isoform expressions in the cells from the benign liver and HCC samples

To identify the mutational gene expressions in each cell, exome sequencing was performed on benign liver, HCC, and gallbladder samples from the same individual. The sequencing results from the gallbladder were used as a germline reference to establish whether the structural variants found in the benign hepatocytes or the HCC samples were somatic mutations. The expression data were then limited to including only non-synonymous mutations detected in the exomes of the benign liver or the HCC samples. The expressions of these mutations were further filtered by the requirement of detecting the same mutation in a minimum of three different cells. Based on these criteria, a total of 2939 mutations were found to be expressed in the HCC and benign hepatocyte samples.

To investigate the role of mutational gene expression in HCC development, mutated gene expression levels were normalized to 'share' of the mutated transcripts relative to all the transcripts of a given gene, while mutational isoform expressions were normalized to 'share' of the mutation isoforms relative to all the transcripts of a given isoform (see nomenclature in 'Methods' for definition). When mutational

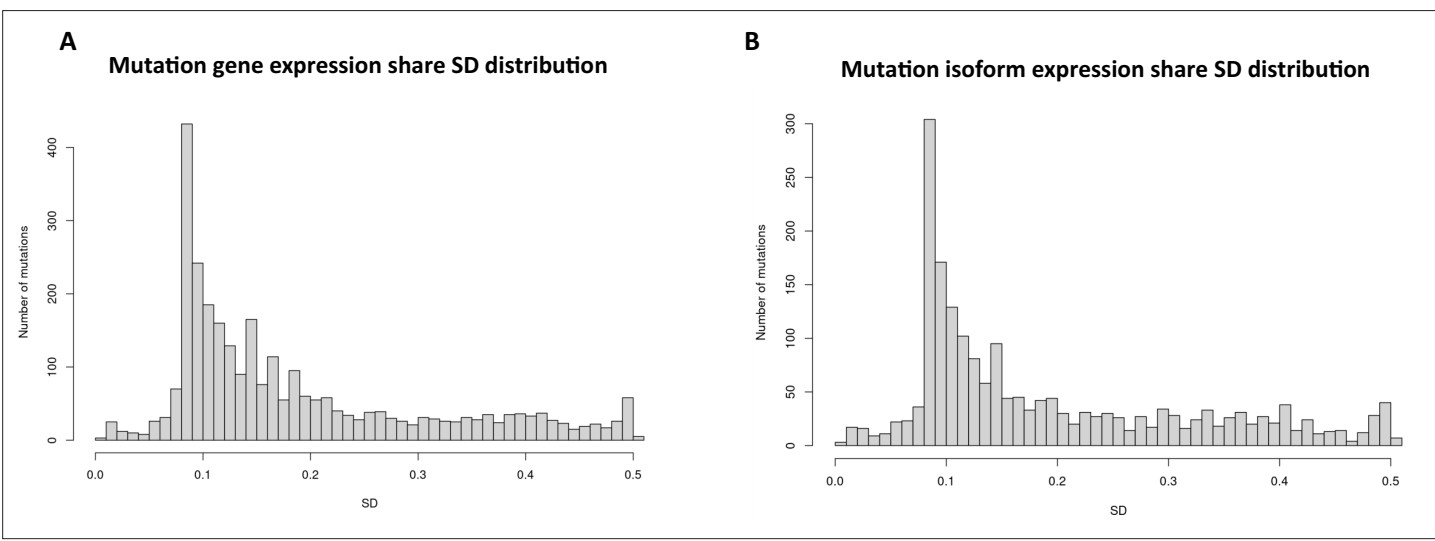

**Figure 2.** Mutation expression standard deviations. (**A**) Mutational gene expressions share standard deviation across all transcriptomes. (**B**) Mutational isoform expression share standard deviation across all transcriptomes.

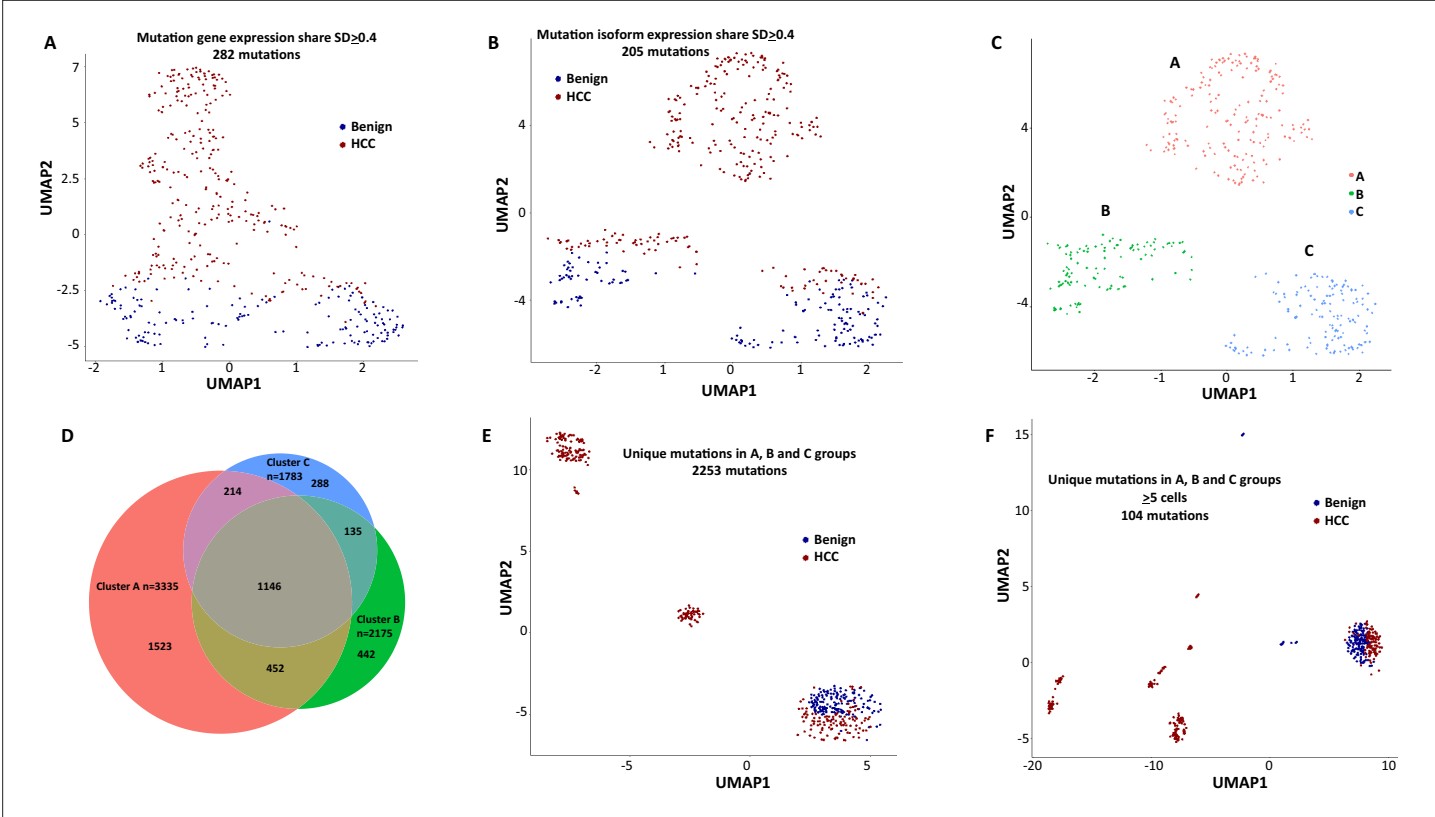

**Figure 3.** Mutation expression clustering of cells from hepatocellular carcinoma (HCC) and its benign liver counterpart. (**A**) Uniform Manifold Approximation and Projection (UMAP) clustering of cells from the HCC and benign liver, based on mutational gene expressions shares with standard deviations ≥ 0.4. Red cells are from HCC; Blue cells are from benign liver. (**B**) UMAP clustering of cells from the HCC and benign liver based on mutational isoform expression shares with standard deviations ≥ 0.4. (**C**) Relabeling of clusters from (**B**) as 'A', 'B', and 'C'. (**D**) Venn diagram of mutational isoform expressions in cells from clusters A, B, and C. (**E**) UMAP clustering of cells from the HCC and benign liver based on the mutational isoform expressions from clusters A, B, and C. (**F**) UMAP clustering of cells from the HCC and benign liver based on the mutational isoform expression in at least five cells from clusters A, B, and C.

The online version of this article includes the following figure supplement(s) for figure 3:

**Figure supplement 1.** Heatmaps of mutational gene expression.

gene expressions were compared across all the transcriptomes, variations of mutated gene expressions were found (*Figure 2A and B*). To achieve an unbiased classification of cells, we chose a knowledge-blind approach to analyze the cell populations in both samples. To remove potential non-contributing mutational gene expressions, genes with a normalized mutated gene expression standard deviation (SD) < 0.4 across all the cells were removed. This resulted in 282 genes with mutated gene expression SDs > 0.4. Uniform Manifold Approximation and Projection (UMAP) analysis was then applied to 447 transcriptomes from the HCC and benign liver samples based on these genes. As shown in *Figure 3A*, *Supplementary file 1*, and *Figure 3—figure supplement 1A*, many cells from the HCC sample clustered to a pole position relative to the cells from the benign liver, while other cells from the HCC sample moved in proximity to the cells from the benign liver. To investigate whether mutational isoform expressions contributed to the development of HCC, similar removal of isoforms with mutated isoform expression SDs < 0.4 was performed. The resulting 205 mutational isoforms were then assessed in cells from the benign liver and HCC samples for UMAP clustering. As shown in *Figure 3B*, *Supplementary file 2*, and *Figure 3—figure supplement 1B*, three distinct clusters emerged: one cluster was entirely composed of cells from the HCC sample, while the other two were mixtures of cells from the benign liver and the HCC samples, suggesting that some of these cells were in the transitional stage.

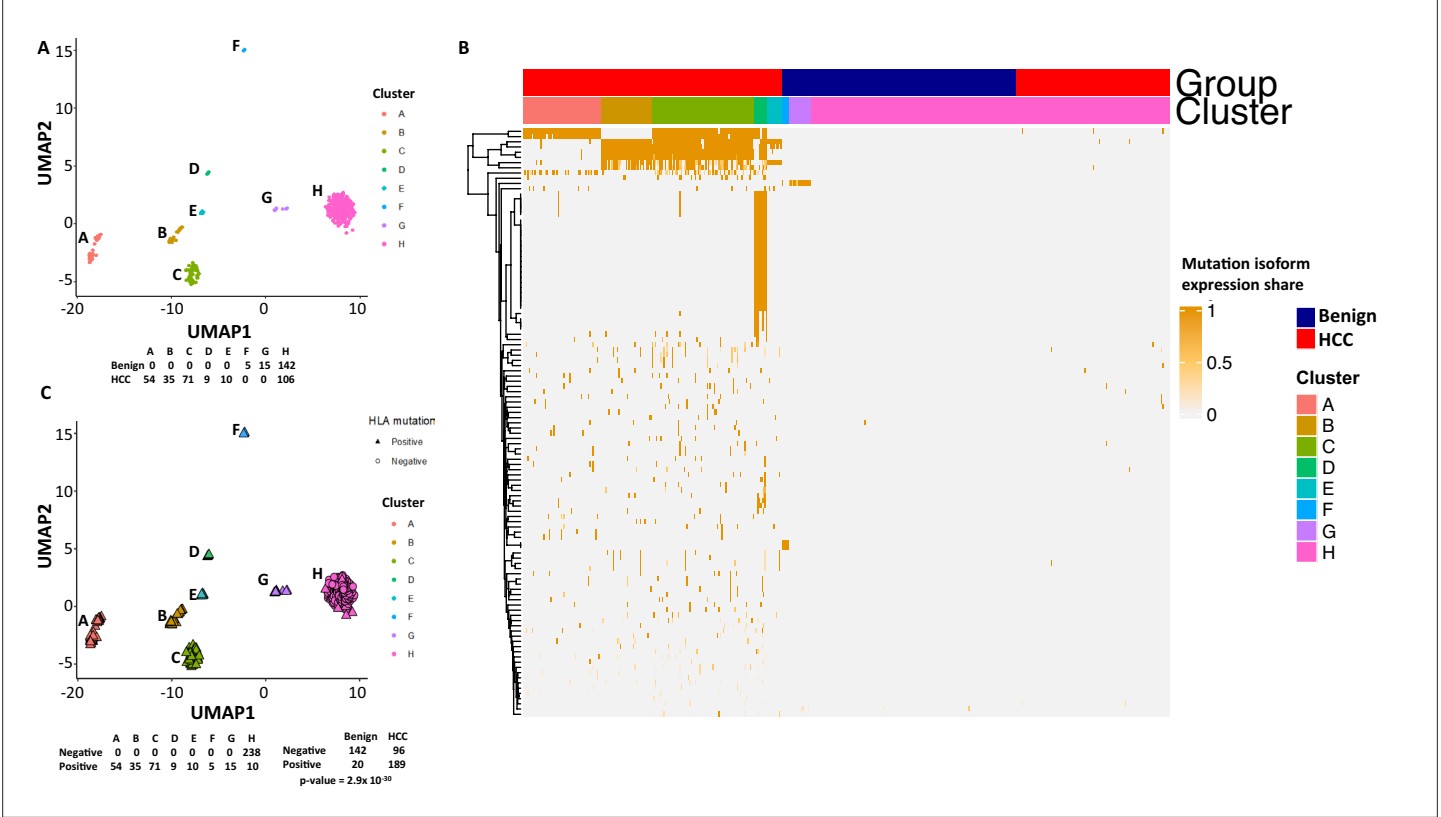

**Figure 4.** Mutations in human leukocyte antigen (HLA) molecules dominated the landscape of hepatocellular carcinoma (HCC) mutational isoform expressions. (**A**) The Uniform Manifold Approximation and Projection (UMAP) clusters from *Figure 2F* are relabeled as A–H groups as indicated. Cells from HCC and benign liver in each cluster are indicated. (**B**) Heatmap of 104 mutational isoform expressions in the HCC and benign liver and clusters A–H. (**C**) Relabeling of UMAP clusters from (**A**), with cells expressing mutation HLA isoforms in triangles. Cells expressing mutation HLA isoforms in each cluster are indicated.

## Mutations of genes involving antigen presentation dominated the mutation expression landscape

To investigate the classifier mutation isoform expression in these transcriptomes that segregate these clusters, the clusters of the mutational isoform expressions in the UMAP were relabeled as A, B, and C groups (*Figure 3C*). A total of 3335 mutation isoforms were found in cluster A, while 2175 and 1783 mutation isoforms were found in clusters B and C, respectively. The overlapping of the mutations from these three groups, as pictured in a Venn diagram (*Figure 3D*), indicated that 1523 mutation isoforms were uniquely present in cluster A, while only 442 and 288 mutations were present in clusters B and C, respectively. To investigate whether these mutation isoform expressions can further classify the cell population, these mutational isoforms were then combined and applied to the cluster analysis of 447 transcriptomes. The UMAP clustering generated four distinct clusters (*Figure 3E*), with three of the clusters composed entirely of cells from the HCC samples, distinctly separated from the fourth cluster, which was a mixture of cells from the benign liver and the HCC. We then limited the mutational isoforms to those that were expressed in at least five single-cell transcriptomes. This uncovered 104 mutations which met the established criteria (*Supplementary file 3*). When UMAP analysis was performed based on these 104 mutation isoforms, eight distinctive clusters were resolved. Cells in all but one clusters co-migrated with cells of their sources (*Figure 3F*).

To examine the mutational isoform expressions in these clusters, eight clusters were relabeled as A–H (*Figure 4A*). Among 104 mutation isoforms, the major histocompatibility complex (human leukocyte antigen [HLA]) was the most prominent, with 68 iterations (60.2%) (*Supplementary file 3*, *Figure 4B*). Specifically, HLA-B NM_005514_2 mutations G283A and C44G were mostly present in cluster A. Cells in cluster B had mutations G572C, G539T, A527T, C463T, and G283A of HLA-B NM_005514_2, and cells in cluster C had mutations G379C and A167T of the same molecule. Cells

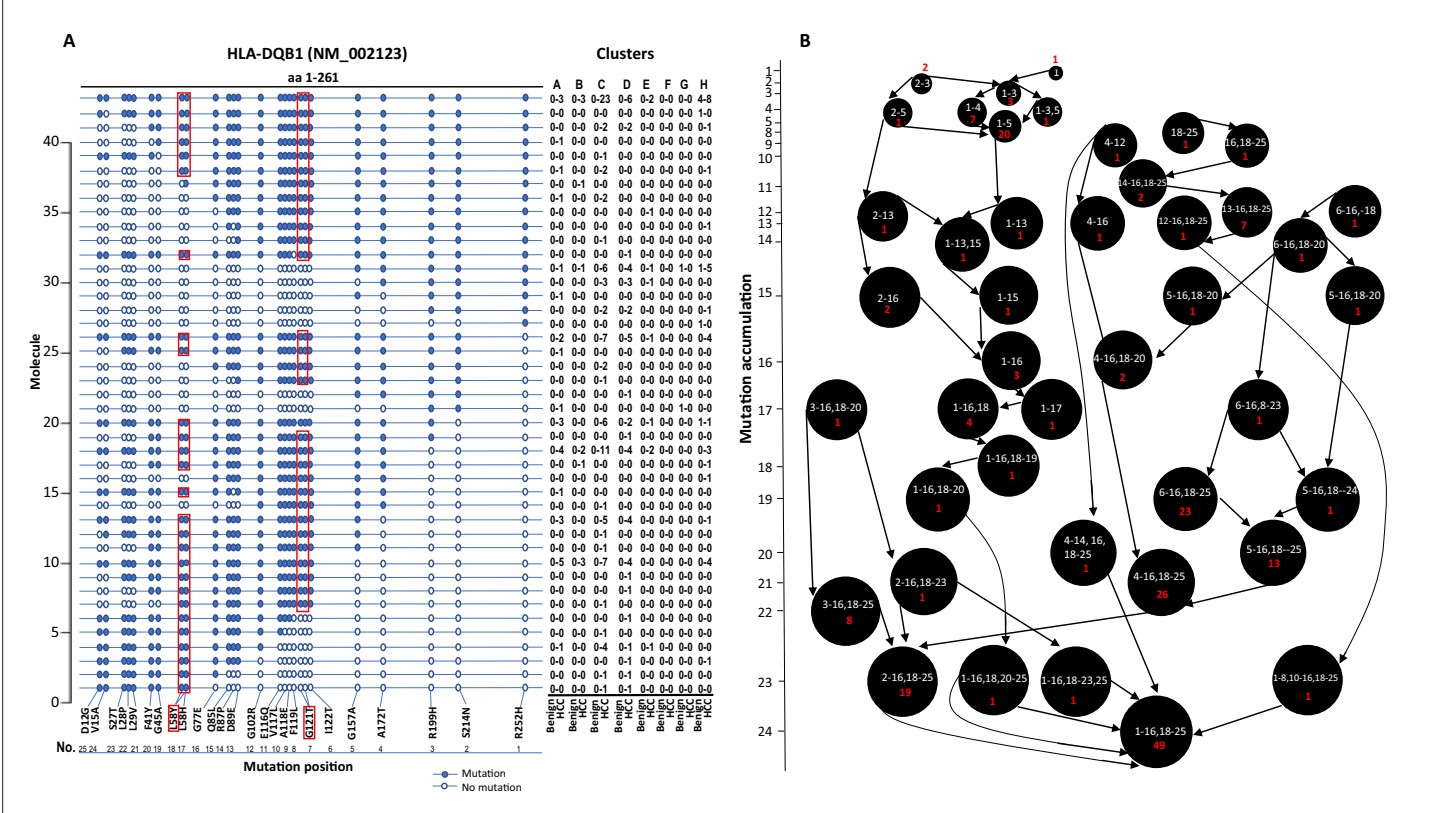

**Figure 5.** Evolution of mutations in HLA-DQB1 molecule. (**A**) Somatic mutations in single molecules of HLA-DQB1 NM_002123. The position of mutation is indicated at the bottom of the graph. The mutation is numerically numbered from C-terminus to N-terminus. The numbers of cells expressing these mutation isoforms from each cluster (indicated at the top) or sample of origin (indicated at the bottom) are shown in the right panel. Close circle, mutation codon; open circle, wild-type codon; open rectangle, double single-nucleotide mutation in the same codon. (**B**) Pathway flowchart of mutation accumulation in single molecules of HLA-DQB1. The area of the circle is proportional to the accumulated number of mutations in a molecule. The scale on the left indicates the number of mutations in a single molecule but is not mathematically scaled. The arrow indicates the potential pathways of mutation accumulation in these molecules. The number of white text indicates specific mutation(s) in a molecule. The number of red text indicates the number of cells expressing the mutation(s).

in cluster D had up to 25 different mutations in HLA-DQB1 NM_002123. Cells in cluster E had partial mutations overlapped with those of clusters A and B. Surprisingly, cells in cluster F, which were from the benign liver, contained unique mutations in HLA-C NM_002117 molecule (T539G, C419T, G176A), while cells in cluster G, another cluster from the benign liver, had mutation G176A in the same molecule in addition to a mutation in ribosomal protein S9 (G525C, RPS9 NM_001013_5). Cluster H was a collection of cells with few mutations in the list. When the clusters were relabeled with mutations in HLA, all cells from clusters A–G were positive for some HLA mutations (*Figure 4C*). On the other hand, only 10 cells from group H were positive for the HLA mutation, suggesting that these mutations in HLA molecules are highly cancer-specific ($2.9 \times 10^{-30}$).

## Evolution of mutations in HLA molecules

Long-read sequencing enabled us to identify multiple mutations in the same molecule. Indeed, most HLA molecules contained multiple mutations. A salient example of a multi-mutation molecule is HLA-DQB1, where up to 25 missense mutations were identified in a single molecule of NM_002123 (*Figure 5A*). We hypothesize that the collection of these mutations started from sporadic isolated mutations and accumulated over time in the development of HCC. To look for the origin of the mutation clusters, we searched for isolated mutation(s) that were the common denominators amongst the larger mutation clusters. We also hypothesize that mutation is irreversible once it occurs. As shown in *Figure 5B*, the largest cluster (49 cells) of mutations occurred in the HLA-DQB1 NM_002123 molecule. The mutation cluster contained 25 single-nucleotide variants that caused 24 amino acid changes

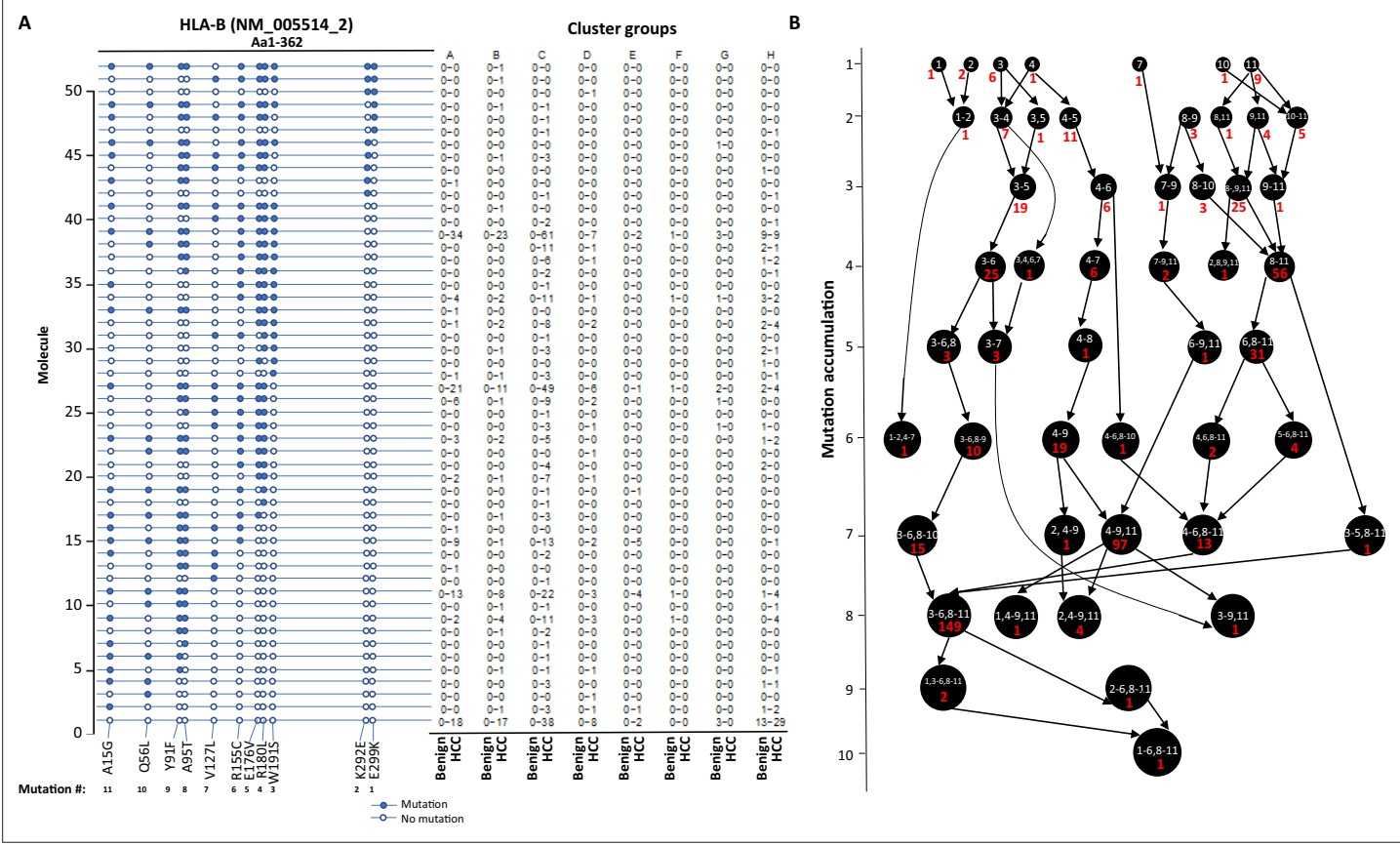

**Figure 6.** Evolution of mutations in HLA-B molecules. (**A**) Somatic mutations in single molecules of HLA-B NM_005514_2. The position of the mutations is indicated at the bottom of the graph. Mutations are numerically numbered from C-terminus to N-terminus. The numbers of cells expressing these mutation transcripts from each cluster or sample are indicated in the right panel. Close circle, mutation codon; open circle, wild-type codon. (**B**) Pathway flowchart of mutation accumulation in single molecules of HLA-B. The area of the circle is proportional to the accumulated number of mutations in a molecule. The scale on the left indicates the number of mutations in a single molecule but is not mathematically scaled. The arrow indicates the potential pathways of mutation accumulation in these molecules. The number in white text indicates the specific mutation(s) in a molecule. The number in red text indicates the number of cells expressing the mutation(s).

within the single molecule. There were several possible nucleotide mutation accumulation pathways that could have led to the formation of this hypermutation cluster. One of the pathways appears to have started at aa252 with the modification of arginine to histidine. The spread of the mutations would have been in one direction from 3' end to 5' end in a mostly contiguous fashion. However, the main pathway of accumulation of mutations is likely to have come from the mid-segment of the molecule since many cells containing subsets of mutations in this segment were detected, albeit they have larger hops in the accumulation process. Some isolated mutations, such as R252H, S214N+R199H, occurred in cells from the benign liver sample. They were associated with malignancy when more mutations were accumulated.

The stepwise accumulation of mutations in single molecules also occurred in HLA-B, HLA-C, and HLA-DRB1. In the HLA-B NM_005514_2 molecule, a total of 11 mutations were identified. The hypermutation cluster in the single protein started from nine different isolated mutations. The main pathway of mutation accumulation appeared to start from the isolated mutations of W191S or A15G (**Figure 6A and B**). These mutations expanded in a contiguous fashion and reached the peak at eight mutations, as evidenced in 149 cells. One cell continued to expand its mutation repertoire up to 10 (**Figure 6B**). For HLA-C NM_002117, 14 different missense mutations were identified (**Figure 7A and B**). The major cascade of the mutation accumulation appeared to start from the isolated mutations of L180R or S140F. The expressions of the combination of L180R and S140F mutations accounted for most cells (n = 222) that contained HLA-C mutants, followed by the combination of L180, S140F, and R59Q (n = 147). One cell accumulated eight mutations in the single molecule (**Figure 7B**). The accumulation of

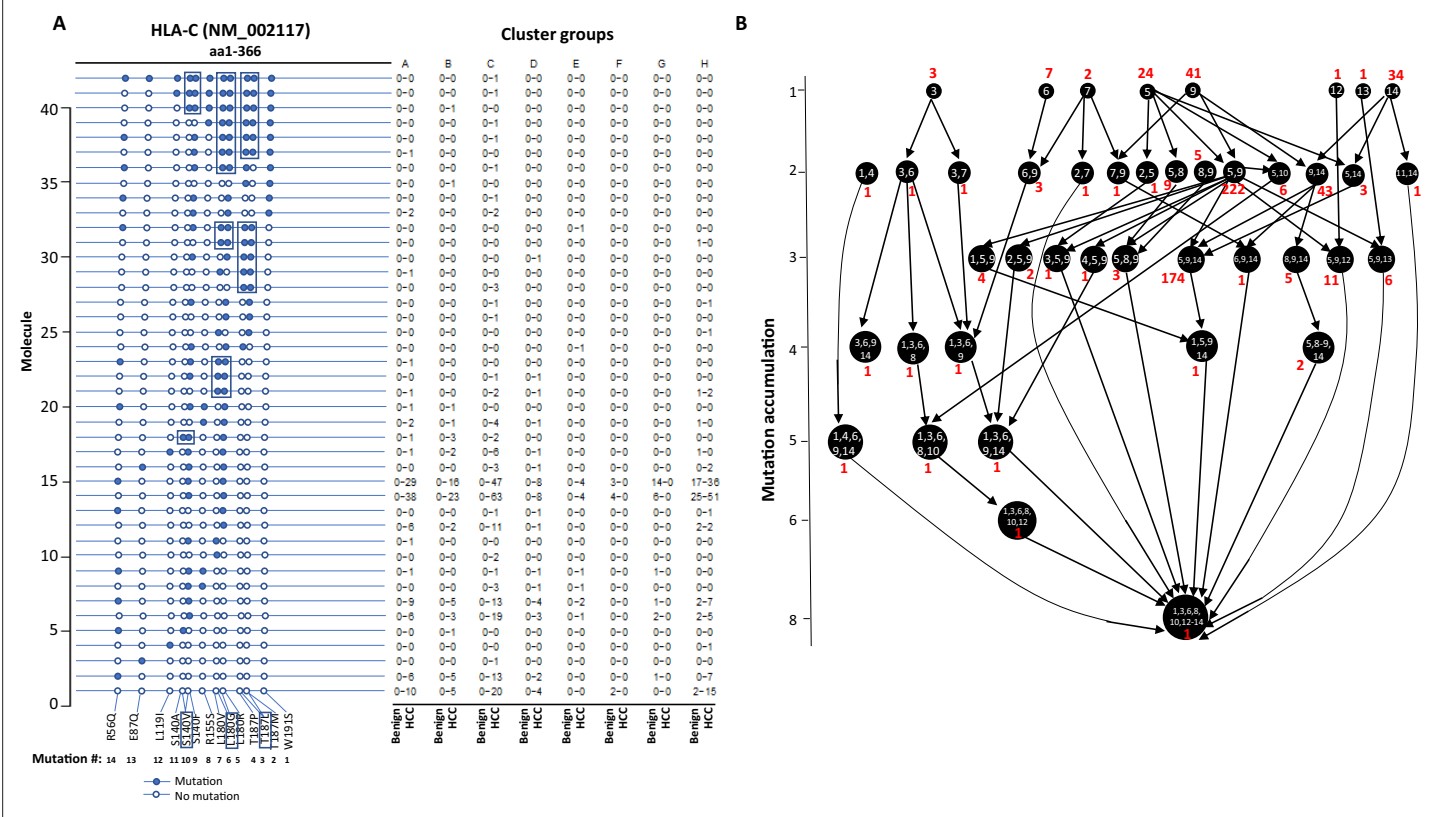

**Figure 7.** Evolution of mutations in HLA-C molecules. (**A**) Somatic mutations in single molecules of HLA-C NM_002117. The position of the mutation is indicated at the bottom of the graph. The mutation is numerically numbered from C-terminus to N-terminus. The numbers of cells expressing these mutation transcripts from each cluster or sample are indicated in the right panel. Close circle, mutation codon; open circle, wild-type codon; open rectangle, double single-nucleotide mutation in the same codon. (**B**) Pathway flowchart of mutation accumulation in single molecules of HLA-C. The area of the circle is proportional to the accumulated number of mutations in a molecule. The scale on the left indicates the number of mutations in a single molecule but is not mathematically scaled. The arrow indicates the potential pathways of mutation accumulation in these molecules. The number in white text indicates specific mutation(s) in a molecule. The number in red text indicates the number of cells expressing the mutation(s).

these mutations appeared non-contiguous. For HLA-DRB1 NM_002124, up to five different mutations in a single molecule were identified (*Figure 8A and B*). All five isolated mutations were identified. The peak mutation accumulation (as seen in 81 cells) is the combination of S133A, A103P, A102G, and T80R. Multiple pathways were detected that might lead to this pattern of mutation accumulation.

## Mutation expression of DOCK8 and STEAP4

DOCK8 is a member of the gene family responsible for guanine nucleotide exchange and has been shown to interact with Rho GTPase (*El Masri and Delon, 2021*; *Harada et al., 2012*). DOCK8 is an important component in intracellular signaling. Isoform expression analysis showed a significant increase in mutation isoform expression in HCC cells (*Supplementary file 3*). Overall, three mutations were detected in DOCK8 (*Figure 9A*). Six types of novel isoforms were detected. Interestingly, no known wild-type isoform expressions were found in either benign liver or HCC cells. All DOCK8 transcripts from HCC cells contained at least one mutation. Some HCC cells expressed two patterns of mutations, suggesting that the mutations occurred in both alleles. The lack of wild-type transcript expression in benign and cancer cells implies that mutation may play a role in DOCK8 expression. On the other hand, STEAP4 is a metalloreductase and carries out the reduction of Fe(3+) to Fe(2+) (*Oosterheert et al., 2018*). STEAP4 was found to contain tumor suppressor activity in several human malignancies (*Tang et al., 2022*; *Wu et al., 2020*; *Zhao et al., 2021*). The G75D mutation appeared to be somatic since this variant was not found in the genome of the gallbladder from the same individual. This mutation is located within the domain of NADP-binding site. Thus, it may impact the enzymatic function of the protein. The expression of STEAP4 transcripts was exclusively found

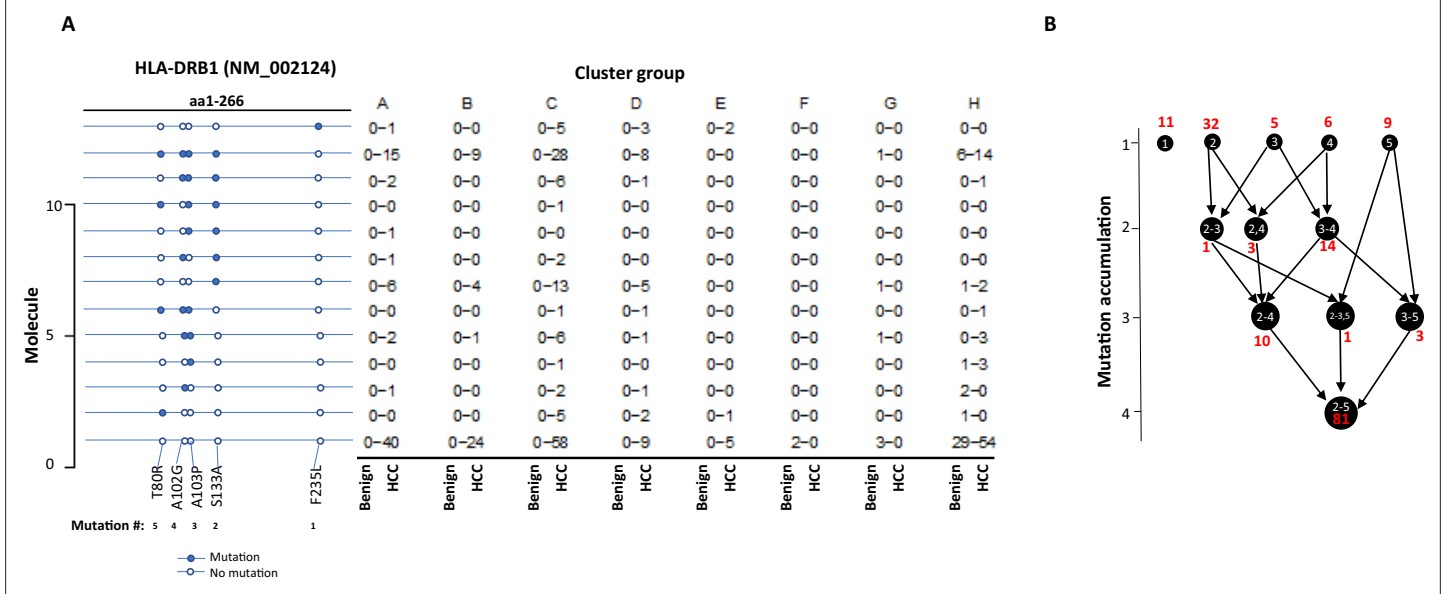

**Figure 8.** Evolution of mutations in HLA-DRB1 molecules. (**A**) Somatic mutations in single molecules of HLA-DRB1 NM_002124. The position of mutation is indicated at the bottom of the graph. The mutation is numerically numbered from C-terminus to N-terminus. The numbers of cells expressing these mutation transcripts from each cluster or sample are indicated in the right panel. Close circle, mutation codon; open circle, wild-type codon. (**B**) Pathway flowchart of mutation accumulation in single molecules of HLA-DRB1. The area of the circle is proportional to the accumulated number of mutations in a molecule. The scale on the left indicated the number of mutations in a single molecule but is not mathematically scaled. The arrow indicates the potential pathways of mutation accumulation in these molecules. The number in white text indicates specific mutation(s) in a molecule. The number in red text indicates the number of cells expressing the mutation(s).

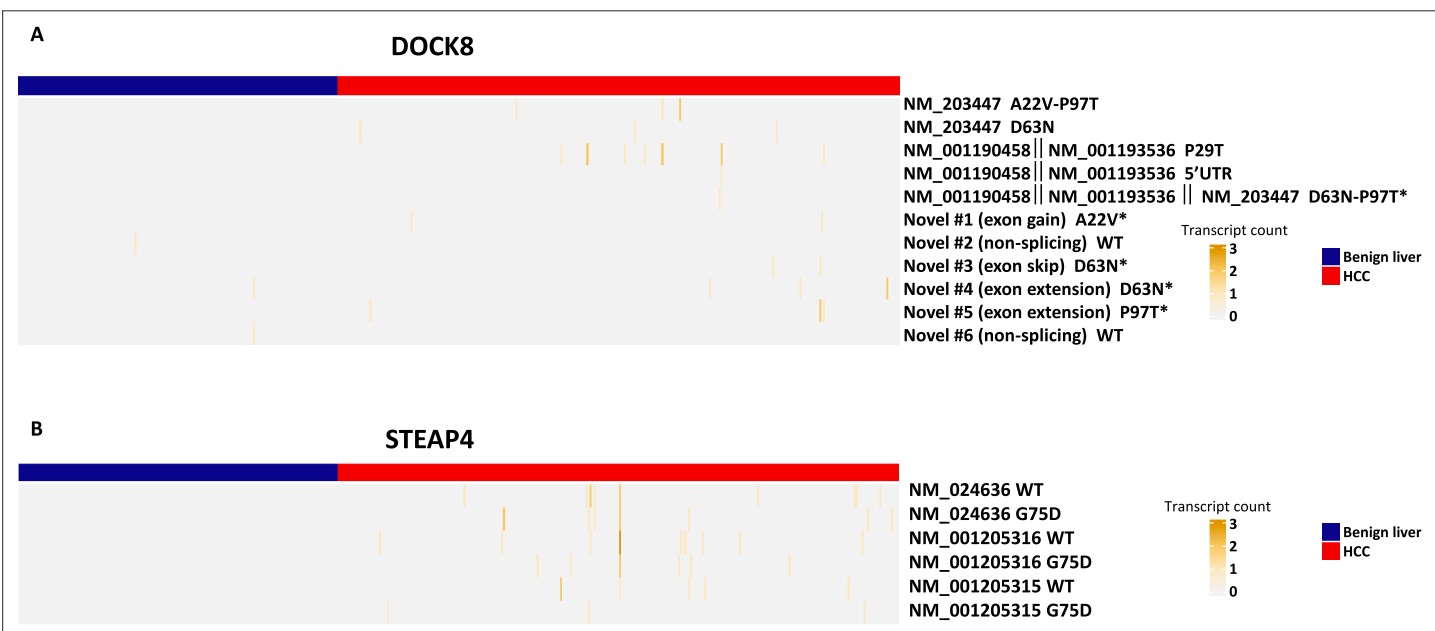

**Figure 9.** Mutation isoform expression of DOCK8 and STEAP4. (**A**) Heatmap of wild-type and mutation isoform expressions of DOCK8. The number of transcript, the position of mutation, and the specific isoforms is indicated. Some transcripts have multiple assignment due to detection of partial transcripts. *Prediciton based on sequence of NM_203447. (**B**) Heatmap of wild-type and mutation isoform expression of STEAP4. The number of transcript, the position of mutation, and the specific isoforms is indicated.

The online version of this article includes the following figure supplement(s) for figure 9:

**Figure supplement 1.** TaqMan RT-PCR of fusion transcripts in hepatocellular carcinoma (HCC) and benign liver samples.

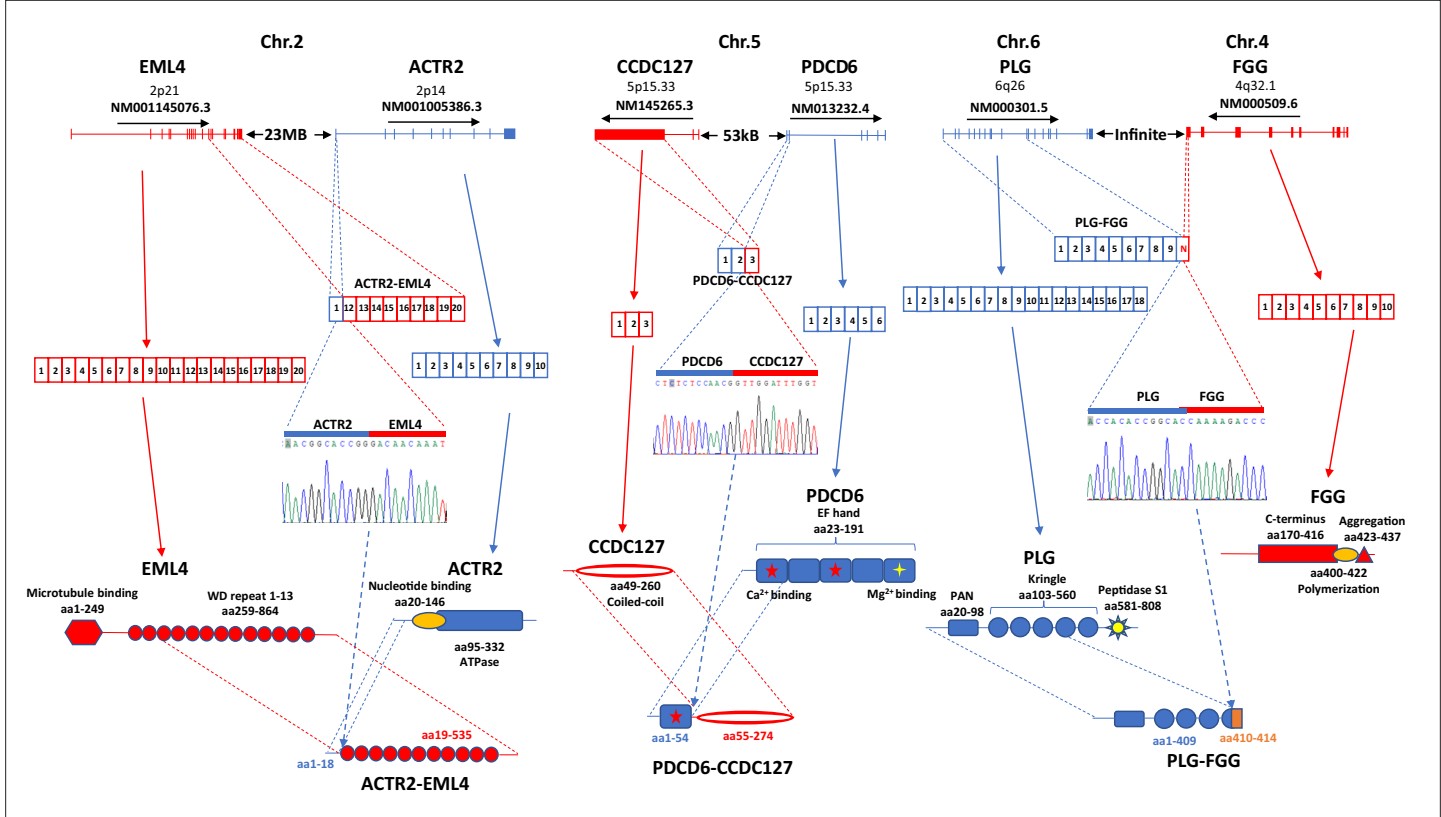

**Figure 10.** Fusion gene expression validation in hepatocellular carcinoma (HCC) sample. Left panel: ACTR2-EML4 fusion. Top: chromosome organization of EML4 and ACTR2 exons. The directions of transcriptions are indicated. Second from the top: exon representations in ACTR2-EML4 fusion transcript, EML4 NM001145076.3, and ACTR2 NM001005386.3. Middle: chromogram of Sanger sequencing. The segments for ACTR2 and EML4 are indicated. Bottom: protein domain and motif organizations of EML4, ACTR2, and ACTR2-EML4 fusion. Middle panel: PDCD6-CCDC127 fusion. Top: chromosome organization of CCDC127 and PDCD6 exons. The directions of transcriptions are indicated. Second from the top: exon representations in PDCD6-CCDC127 fusion transcript, CCDC127 NM145265.3, and PDCD6 NM013232.4. Middle: chromogram of Sanger sequencing. The segments for PDCD6 and CCDC127 are indicated. Bottom: protein domain and motif organizations of CCDC127, PDCD6, and PDCD6-CCDC127 fusion. Right panel: PLG-FGG fusion. Top: chromosome organization of PLG and FGG exons. The directions of transcriptions are indicated. Second from the top: exon representations in PLG-FGG fusion transcript, PLG NM000301.5, and FGG NM000509.6. Middle: chromogram of Sanger sequencing. The segments for PLG and FGG are indicated. Bottom: protein domain and motif organizations of PLG, FGG, and PLG-FGG fusion. The open-reading frame of FGG was eliminated due to frameshift in PLG-FGG fusion. Unrelated four additional amino acids were added to the truncated N-terminus of PLG.

in HCC cells (*Figure 9B*). Both wild-type and mutant STEAP4 transcripts were identified. One HCC cell expressed mutant and wild-type transcripts in multiple isoforms. In contrast to DOCK8, STEAP4 mutation expression was not biased.

## Fusion gene expression in single-cell level

Gene fusion is one of the hallmarks of human cancers. To identify fusion gene transcripts in the sample, we applied SQANTI (*Tardaguila et al., 2018*) annotation to the long-reads in order to identify transcripts that mapped to two different genes using the criteria described previously (*Liu et al., 2021a*; *Yu et al., 2019a*; *Yu et al., 2014a*) and in the methods. To rule out potential artificial chimera, the fusion gene must be corroborated by at least two different cells. After multilayer screening, 21 fusion genes were identified, and 3 fusion genes were selected to validate experimentally. Among these fusion genes, ACTR2-EML4 was detected only in the cancer sample (*Figure 10* and *Figure 9—figure supplement 1*). ACTR2 is a major component of ARP2/3 complex and is responsible for cell shape and motility, while EML4 contains WD repeats that are essential for protein-protein interaction. The fusion retains most of the WD repeat domain from EML4 while removing most of the amino acid sequence from ACTR2. The loss of microtubule-binding domain may negatively impact the microtubule organization activity of the EML4 domain of the fusion protein. PDCD6 is an EF-hand domain-containing

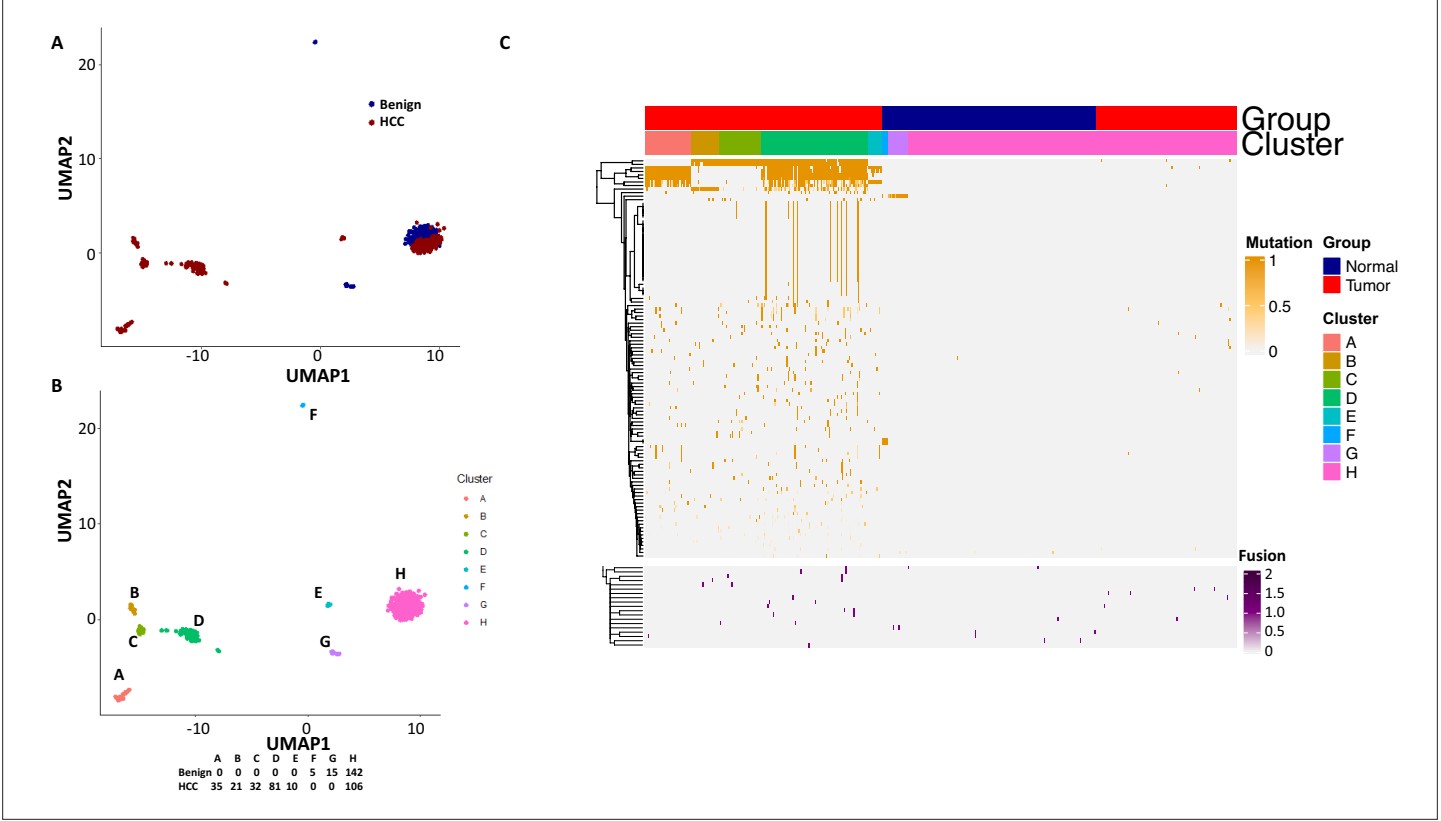

**Figure 11.** The impact of fusion gene expressions on the cell clustering generated by mutational isoform expressions. (**A**) Uniform Manifold Approximation and Projection (UMAP) cluster analysis of cells from hepatocellular carcinoma (HCC) and benign liver based on 104 mutational isoform expressions and 20 fusion gene expressions. (**B**) Relabeling of clusters from (**A**) as clusters A–H. (**C**) Heatmap of mutational isoform expressions and fusion gene expression shares for clusters A–H. The cells from HCC and benign liver are indicated.

protein and has calcium and magnesium-binding activity. CCDC127 is a coiled-coil domain containing transcription repressor. The PDCD6-CCDC127 fusion retained most of the coiled-coil domain from CCDC127 and a single EF-hand domain from PDCD6. The signaling response of the fusion protein may be altered because of the new calcium-binding motif in the molecule. Finally, the FLG-FGG fusion is a unique chromosomal translocation product where the chromosome breakpoint is located in the exons. The fusion is a truncation of plasminogen. The removal of the C-terminus from plasminogen may lead to constitutive activation of its protease and enhance blood coagulation and other cell signaling activities of plasminogen.

To investigate whether fusion transcripts had an impact on transcriptome clusters, we added these fusion genes to the mix of 104 mutational isoforms to perform UMAP analysis. As shown in *Figure 11A–C*, the cancer cell clusters A–D appeared to shift significantly to the left and underwent major reshuffling among the groups. On the other hand, clusters F–H remained in similar positions, while cluster E moved to the right, indicating that fusions impacted mostly the characteristics of cancer cells but had a very limited impact on benign hepatocytes.

## Mutational gene expression and fusion transcript enhanced transcriptome clustering of benign hepatocytes and HCC

Cell clustering and segregation can be determined by the differential expression of transcripts. Our mutational gene expression analyses suggested that some benign hepatocytes harbored mutations that resembled those of malignant cells. To reduce the complexity amongst the transcriptomes, we removed genes or isoforms across all samples with expression SDs < 0.5, 0.8, 1.0, and 1.4, respectively. As shown in *Figure 12*, *Figure 12—figure supplements 1–8*, and *Supplementary files 4–11*, the segregation of two groups of cells occurred when genes or isoforms had SDs > 0.5. The segregation became more pronounced when the SDs were larger, with mostly malignant cells in one group

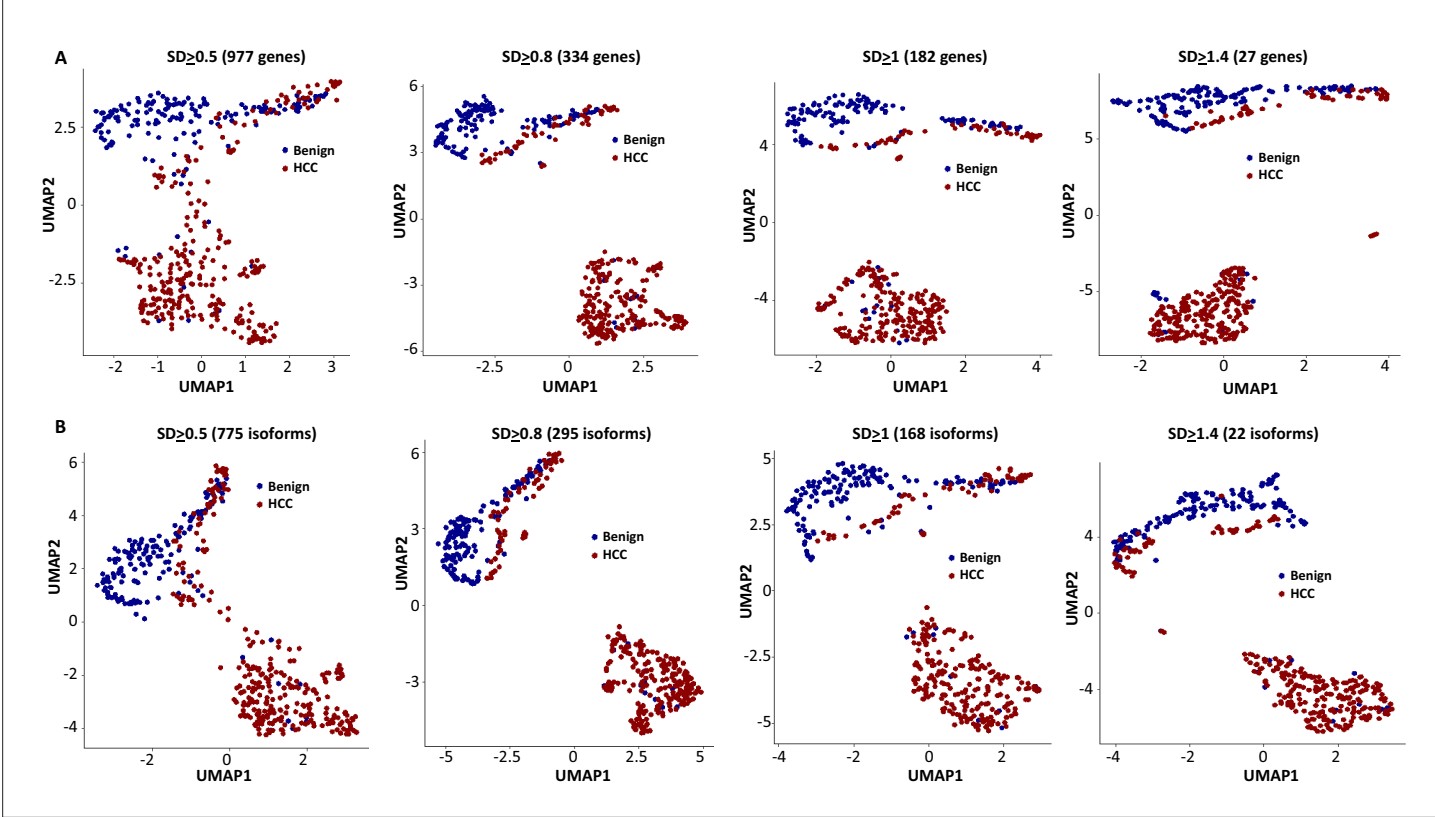

**Figure 12.** Uniform Manifold Approximation and Projection (UMAP) cluster analyses of cells from the hepatocellular carcinoma (HCC) and benign liver based on gene expressions with different standard deviation cutoffs. (**A**) UMAP clustering of cells based on gene expression with standard deviations at least 0.5, 0.8, 1.0, or 1.4. The numbers of genes employed in the UMAP analysis are indicated. Blue dot, cell from the benign liver; red dot, cell from HCC. (**B**) UMAP clustering of cells based on isoform expressions with standard deviations at least 0.5, 0.8, 1.0, or 1.4. The numbers of genes employed in the UMAP analysis are indicated. Blue dot, cell from the benign liver; red dot, cell from HCC.

The online version of this article includes the following figure supplement(s) for figure 12:

**Figure supplement 1.** Segregation of cells between hepatocellular carcinoma (HCC) and benign liver samples.

**Figure supplement 2.** Segregation of cells between hepatocellular carcinoma (HCC) and benign liver samples.

**Figure supplement 3.** Segregation of cells between hepatocellular carcinoma (HCC) and benign liver samples.

**Figure supplement 4.** Segregation of cells between hepatocellular carcinoma (HCC) and benign liver samples.

**Figure supplement 5.** Segregation of cells between hepatocellular carcinoma (HCC) and benign liver samples.

**Figure supplement 6.** Segregation of cells between hepatocellular carcinoma (HCC) and benign liver samples.

**Figure supplement 7.** Segregation of cells between hepatocellular carcinoma (HCC) and benign liver samples.

**Figure supplement 8.** Segregation of cells between hepatocellular carcinoma (HCC) and benign liver samples.

and a mixture of malignant and benign hepatocytes in the other. Such cluster segregations were similarly found in either genes or isoform analyses. To examine the relationship between the isoforms and genes, the lists of isoforms and genes at each SD were overlapped through Venn diagrams (**Figure 13A–D**). Interestingly, gene lists included all the isoforms within the same range of SD. To investigate the roles of gene expression alterations that were not accompanied with isoform expression changes, UMAP analyses were performed based on the non-overlapped genes. The results indicated a dramatic reduction of segregation of cells between benign liver and HCC. In contrast, gene-based clustering using genes that showed both gene and isoform-level changes had segregations between benign hepatocytes and HCC cells similar to those performed with the full lists, suggesting that the isoform alterations were the underlying causes that separated the cells between these two samples. Examination of the gene list (182, **Supplementary file 6**) with SDs ≥ 1.0 showed a consistent down-expression of genes of apolipoprotein family, up-expression of genes of ribosomal

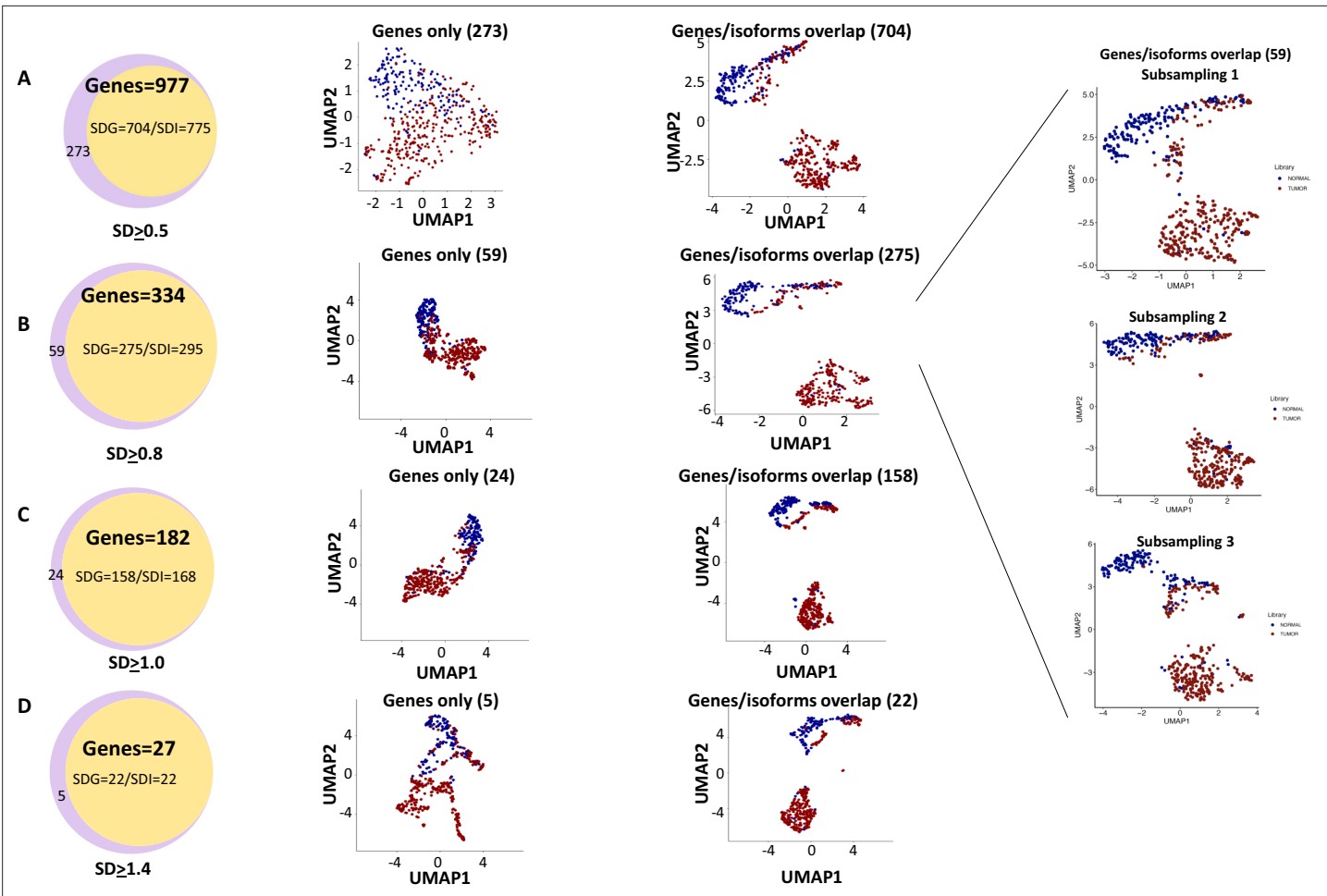

**Figure 13.** Genes with isoform expression alterations played key roles in segregating cells between the hepatocellular carcinoma (HCC) and benign liver samples. (**A**) The role of isoform expressions in segregating cells between the HCC and benign liver when the standard deviation was ≥0.5. Left panel: Venn diagram between gene expressions and isoform expressions with standard deviations ≥ 0.5. Middle panel: Uniform Manifold Approximation and Projection (UMAP) clustering with genes not overlapping with isoforms. Right panel: UMAP clustering with genes overlapping with isoforms. (**B**) The role of isoform expressions in segregating cells between the HCC and benign liver when the standard deviation was ≥0.8. Left panel: Venn diagram between gene expressions and isoform expressions with standard deviations ≥ 0.8. Middle panel: UMAP clustering with genes not overlapping with isoforms. Right panel: UMAP clustering with genes overlapping with isoforms. Blowup panel: UMAP clustering with genes overlapping with isoforms in three subsampling trials. (**C**) The role of isoform expression in segregating cells between the HCC and benign liver when the standard deviation ≥ 1.0. Left panel: Venn diagram between gene expressions and isoform expressions with standard deviations ≥ 1.0. Middle panel: UMAP clustering with genes not overlapping with isoforms. Right panel: UMAP clustering with genes overlapping with isoforms. (**D**) The role of isoform expression in segregating cells between the HCC and benign liver when the standard deviation was ≥1.4. Left panel: Venn diagram between gene expressions and isoform expressions with a standard deviation ≥ 1.4. Middle panel: UMAP clustering with genes not overlapping with isoforms. Right panel: UMAP clustering with genes overlapping with isoforms.

protein family and HLA family in cells from the HCC sample, indicating that cancer cells were less hepatic differentiated but more active in protein synthesis and immune evasion. These abnormalities appeared similar to previous gene expression analyses in both single-cell and bulk sample levels (*Liu et al., 2022*; *Liu et al., 2021b*; *Luo et al., 2006*; *Ng et al., 2021*).

To investigate whether the mutation analysis improved the segregation between cells from the benign liver and HCC, UMAP analysis was performed using gene expressions with SDs ≥ 1.0 (182 non-mutated genes) and gene mutation expression with SDs > 0.4 (282 mutated genes). The results showed that the combination of gene and mutational gene expressions generated three clusters: with two clusters comprised mostly cells from the cancer sample and one cluster of cells mostly from the benign liver (*Figure 14A–C*). When the clusters were relabeled as A, B, and C, cluster A (mostly benign hepatocyte group) had a gain of 7 cells from the benign liver sample and loss of 27 cells from the HCC

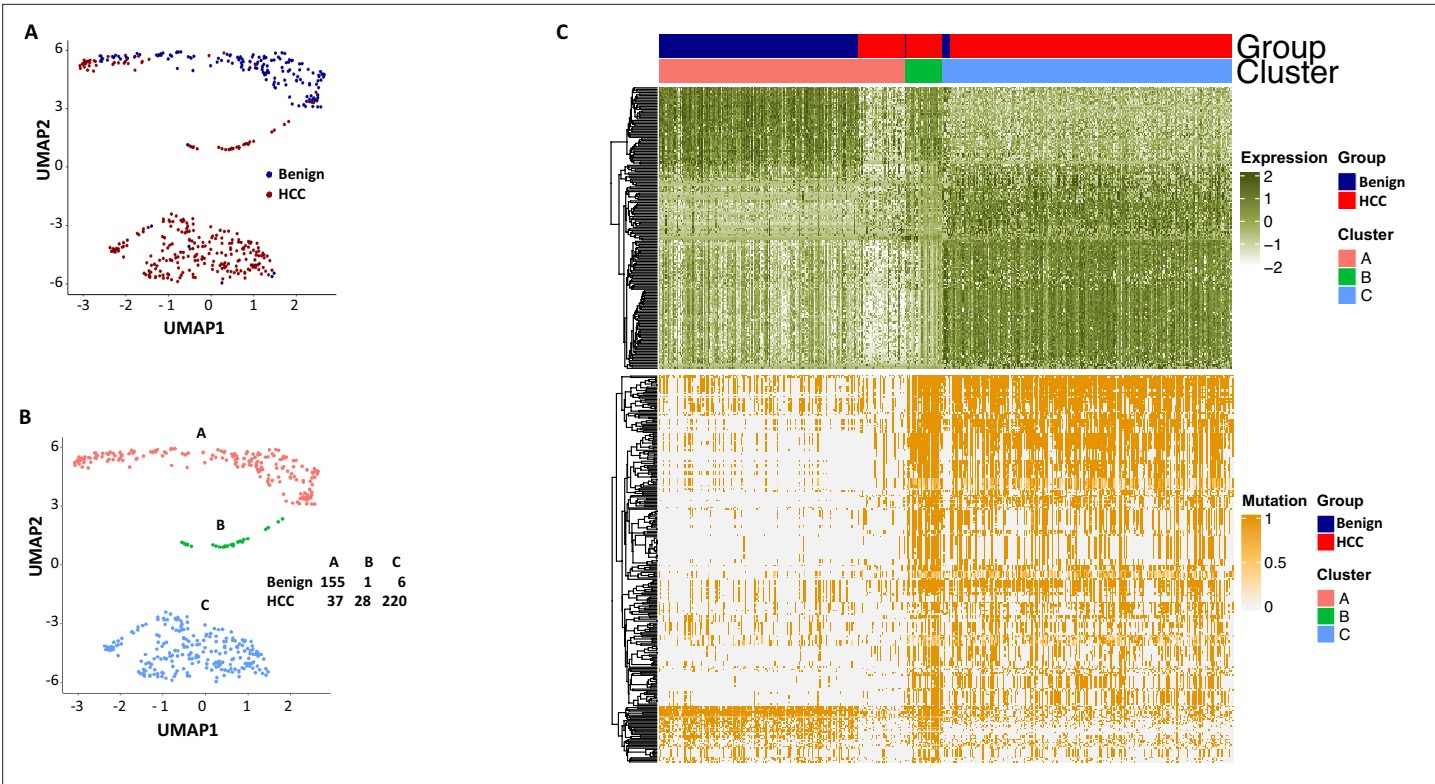

**Figure 14.** Uniform Manifold Approximation and Projection (UMAP) clustering of cells from hepatocellular carcinoma (HCC) and benign liver based on the combination of normal gene expressions and mutational gene expression shares. (**A**) UMAP clustering of cells from the HCC and benign liver samples based on 182 gene expressions with standard deviations ≥ 1.0 and 282 mutational gene expression shares with standard deviations ≥ 0.4, (**B**) Relabeling of clusters from (**A**) as clusters A, B, and C. The numbers of cells from the HCC and benign liver in each cluster are indicated. (**C**) Heatmap of 182 gene expressions and 282 shares for cells from clusters A, B, and C. Cells from the HCC and benign liver mutational gene expressions are indicated.

sample in comparison with that of gene expression analysis alone (*Figure 12—figure supplement 3*), suggesting that the mutation analysis helped to reclassify some of the cells misassigned by gene expression analysis. To investigate whether fusion gene analysis would add value to the clustering of cells from HCC and benign liver, fusion genes were added to the UMAP analysis. The results showed that cluster B moved closer to the cancer cell cluster (cluster C, *Figure 15A–C* and *Supplementary file 12*). Cluster A gained one cell from the benign liver sample and four cells from the HCC sample. Five cells from the benign liver were consistently classified as cancer by the cluster analysis. Further analysis showed that these cells had significant down-expression of genes of apolipoprotein family, up-expression of genes of ribosomal protein and HLA families, and extensive mutations in HLA molecules, suggesting that they were probably the cancer cells embedded in the benign liver sample. The pathway analyses based on gene expression showed that genes in the eukaryotic initiation factor signaling pathway dominated the differential expression gene list with a -log p-value of 104, followed by acute phase response signaling pathway (-log p=35.8) (*Supplementary file 13*). EIF2 pathway is essential for protein translation and thus cell survival and proliferation and were shown to be misregulated in a variety of human cancer (*Silvera et al., 2010*), while acute phase response signaling may play a role in anti-apoptotic activity of cancer cells and signals a poor clinical outcome (*Janciauskiene et al., 2021*). The dysregulation of these pathways appeared to underlie the molecular mechanisms of HCC carcinogenesis.

## Discussion

Long-read sequencing is essential to detect isoform expressions from genes. Synthetic long-read sequencing offers a valuable solution to analyze isoform transcripts of a gene because of its high

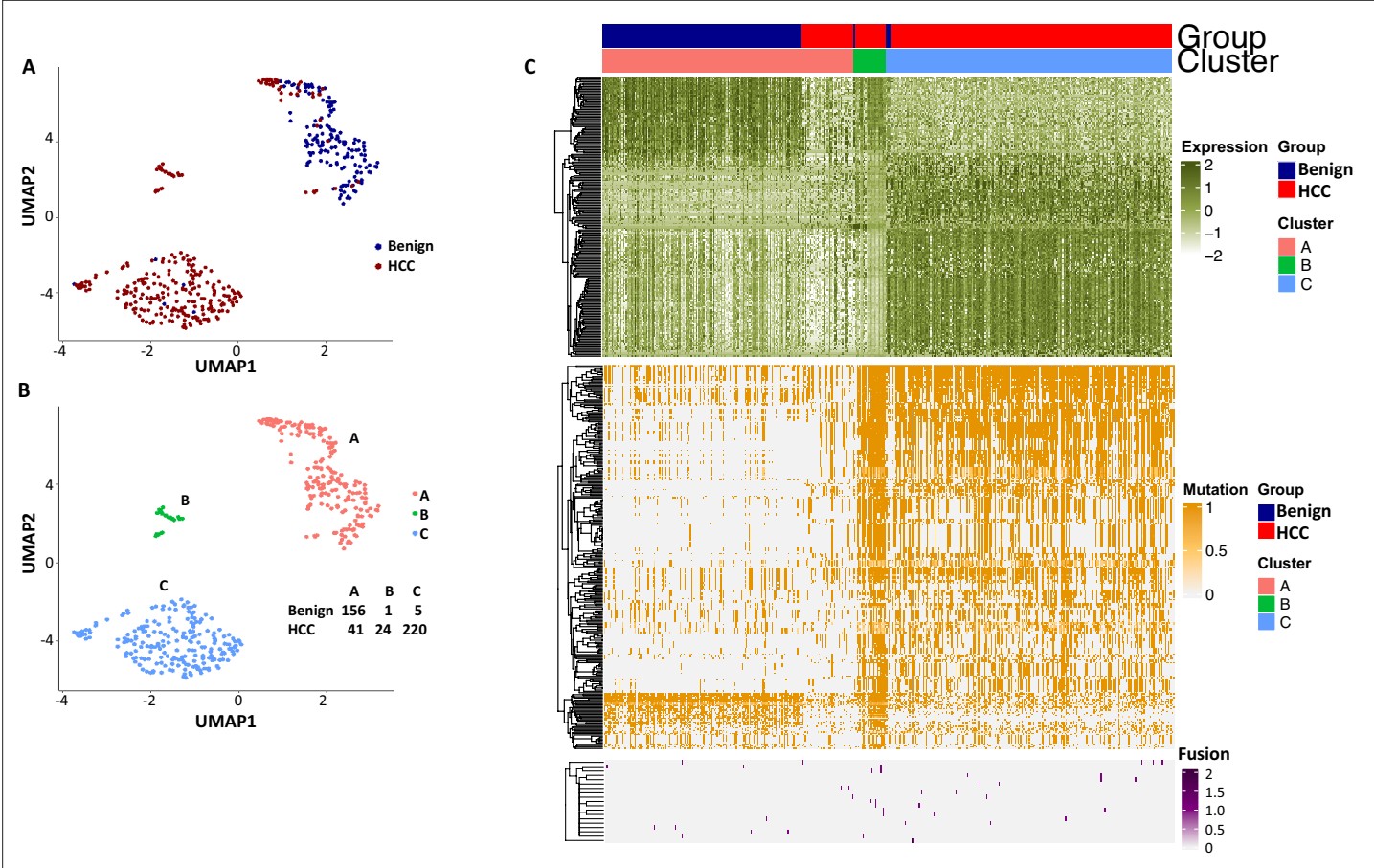

**Figure 15.** Uniform Manifold Approximation and Projection (UMAP) clustering of cells from the hepatocellular carcinoma (HCC) and benign liver based on the combination of normal gene expression, mutational gene expression share, and fusion gene expression share. (**A**) UMAP clustering of cells from HCC and benign liver samples based on 182 gene expressions with a standard deviation ≥ 1, 282 mutational gene expression shares with standard deviations ≥ 0.4, and 20 fusion gene expression shares of any standard deviation. (**B**) Relabeling of clusters from (**A**) as clusters 'A', 'B', and 'C'. The number of cells from HCC and benign liver in each cluster is indicated. (**C**) Heatmap of 182 gene expressions, 282 mutational gene expression shares, and 20 fusion genes expression shares for cells from clusters 'A', 'B', and 'C'. Cells from the HCC and benign liver are indicated.

accuracy, low-error rate, and quantification suitability. However, due to the low-yield nature of most synthetic long-read sequencing methodologies for transcriptome analysis, analyses are mostly limited to a few targeted genes (*Gupta et al., 2018*). To our knowledge, this is the first study to analyze broad-spectrum mutational isoform expression at the single-cell level using synthetic long-read sequencing technology. The technology described in this study may have broad utility in biology and medicine: it can be applied to quantify the diversity of isoform expression, resolve mutational gene expression, and be used to discover novel fusion genes and new isoforms in any mammalian biological system. For medical research, the technology may help determine which specific protein structure should be targeted by making the specific mutational isoform expression information available.

Currently, there is a lack of studies on multiple mutations in a single-molecule or mutational gene expression at the single-cell level due to the absence of a reliable method. In comparison with previous long-read studies in HCC, which were largely limited to splicing abnormality analysis (*Chen et al., 2019*; *Kiyose et al., 2022*; *van Buuren et al., 2022*), our study suggested that gene expressions were mainly dominant by one or two specific isoforms for a given cell (*Figure 16*). The expressions of HLA-B, -C, -DQB1, and -DRB1, regardless of mutation status, were restricted to one known isoform. Currently, it is still unclear what the translation efficiency of these mutation isoform is. The preference of specific isoform expression in a given cell may be due to the preferred splicing process of cells of different lineages or of varying differentiation stages. Interestingly, mutational gene expression of antigen-presenting genes dominated the expressed mutation list from HCC cells. Most mutations

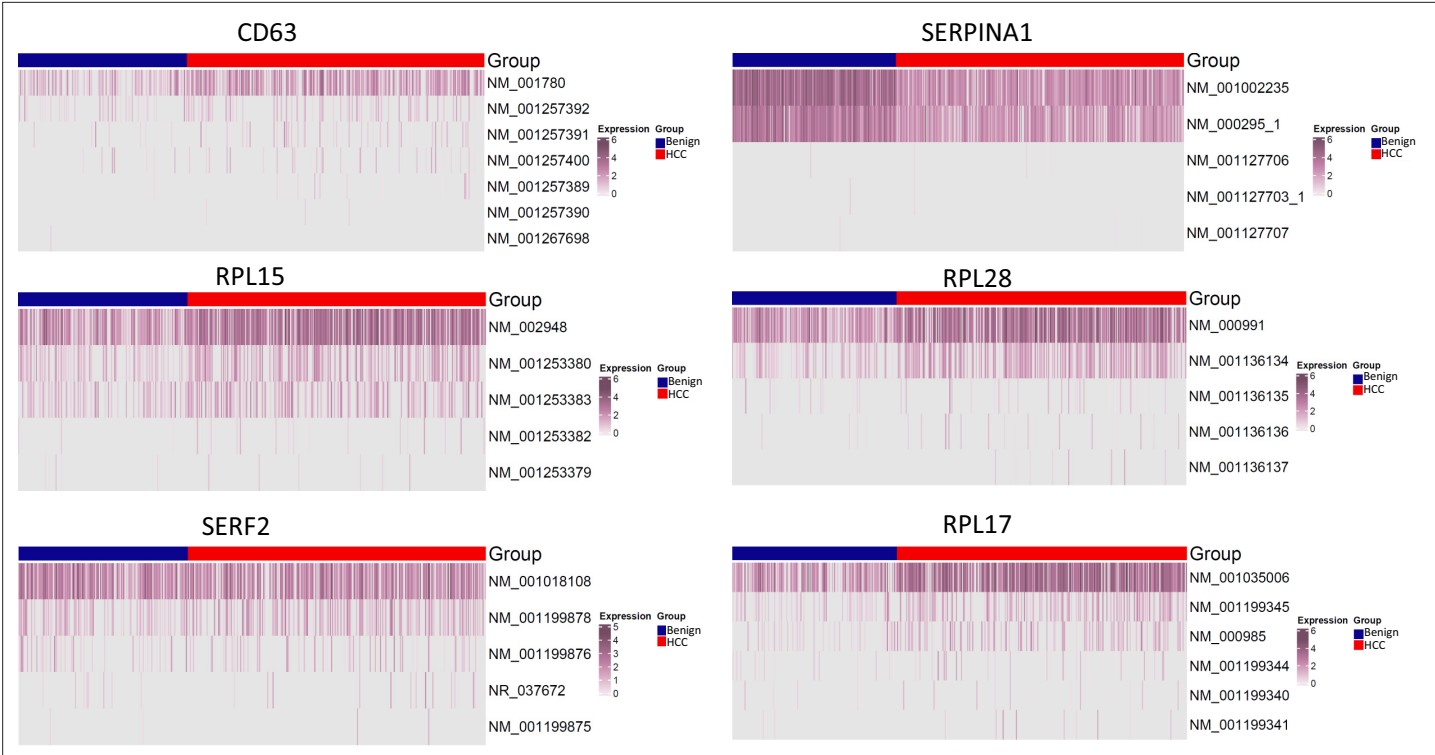

**Figure 16.** Isoform distribution of CD63, SERPINA1, RPL10, RPL28, SERF2, and RPL17. The expression levels of known isoforms are shown.

occurred in the extracellular domain of the HLA molecules. For HLA-B and HLA-C, all three α-domains were mutated, and for HLA-DQB1 and HLA-DRB1, both β-domains and the peptide-binding motifs were impacted. These mutations may alter the interaction with T lymphocytes (*Chan et al., 2018*; *Kondo et al., 2004*). There was a broad spectrum of somatic mutations that affected the HLA gene since both cytosolic and endocytic pathways of antigen presentation may be blocked (*Arnaiz-Villena et al., 2022*; *Manoury et al., 2022*). Interestingly, the expression levels of the mutated HLA molecules also increased in comparison with wild-type alleles from the benign hepatocytes. The hypermutations of these HLA molecules may shield cancer cells from being recognized and targeted by T lymphocytes and allow the cancer cells to evade the host immune surveillance.

The hypermutations in several HLA molecules are of interest because they probably did not happen overnight. Several isolated mutations were also detected in cells from the benign liver samples, suggesting that these mutations accumulated through a clonal progression fashion from a relatively benign background. In the process of malignant transformation, additional mutations in the HLA molecules were acquired due to the pressure from the cellular immune response. Malignant cells with few mutations in the HLA molecules may be destroyed by T lymphocytes, while those with newer mutations evaded the attack. Presumably, the cellular immune system adapted to the new mutations of these HLA molecules and resumed the response to the cancer cells. These cycles may continue to the extent that the mutations overwhelmed the cellular immune system. However, such hyper-mutation clusters may make cancer cells highly vulnerable to artificial immune intervention such as drug conjugated (*Thomas et al., 2016*) or radio-isotope (*Bush, 2002*; *Guleria et al., 2017*) labeled humanized antibody specific for these mutations or cancer vaccine since almost all of these mutations are in the extracellular domains. CRISPR-cas9-mediated genome targeting (*Chen et al., 2017*) at these mutation sites could be an option.

## Methods

### Nomenclature

Mutation gene expression share is defined as a mutated transcript fraction of all the transcripts of a given gene. For example, if 50 transcripts of gene α are detected and 10 of these transcripts are mutated, the mutation gene expression share for gene α is 10/50 = 0.2. Mutation isoform expression share is defined as a mutated isoform fraction of all the transcripts of a given isoform. For example, if 30 transcripts of gene α are A isoform and 20 transcripts B isoform, and 10 mutated transcripts are A isoform transcripts, the mutation isoform expression share for A isoform is 10/30 = 0.3, and for B isoform 0/20 = 0. Fusion gene expression share is defined as a fusion transcript fraction of all the transcripts from head gene and tail gene. For example, if 5 transcripts of fusion gene κ are detected while 10 head gene and 15 tail gene transcripts are also found, the fusion gene expression share is 5/(5 + 10 + 15) = 0.167.

### Single-cell sample preparation

HCC samples and benign liver samples were freshly dissected from a patient who underwent liver transplantation. The procurement procedure was approved by the institution review board of University of Pittsburgh. The procedure was compliant with all regulations related to the protocol (Study# 19030097). The dissected tissues were minced by scalpel and digested with collagenase/protease solution (VitaCyte, 007-1010) until the tissue was fully digested. The digestion time for each preparation was in a range 45–60 min. The digested tissue was removed and immediately cooled with ice-cold Leibovitz's L-15 Medium (Invitrogen, 11415114) supplemented with 10% fetal bovine serum (Sigma, F4135). The single-cell suspension was verified under the microscope. The number of live cells was estimated by trypan blue staining using a hemacytometer.

### 10x Genomics single-cell and UMI barcoding

Approximately 200–300 cells from both HCC or benign liver samples were loaded onto the Chromium next GEM chip G, where the cells were encapsulated with oligo-dT-coated Gel Beads and partitioning oil. The Gel Beads are subsequently dissolved, and the individual cells are lysed. Using the Chromium Next GEM Single Cell 3′Reagent Kit v3.1 from 10x Genomics, Inc, first-strand synthesis was performed using the following thermal cycler parameters: with the lid set at 53°C, incubate at 53°C for 45 min, followed by 85°C for 5 min. The first-strand cDNA was then purified using the kit provided Dynabead clean-up mix. cDNA was then amplified using the provided primers using the following program: with the lid set at 105°C, 98°C for 3 min, then 11 cycles of 98°C for 15 s, 63°C for 20 s, 72°C for 1 min, ended with 72°C for 1 min. Samples were pooled by group prior to long-read library preparation.

### LoopSeq UMI ligation and transcriptome enrichment

The 10X Genomics barcoded cDNA were appended with a LoopSeq-specific adapter (containing the LoopSeq UMI) using a one-step barcoding method. Then, 4 μL of water, 11 μL of Barcoding Master Mix, and 5 μL of 10 ng of single-cell cDNA were combined. The 20 μL reaction is incubated with a 100°C heated lid at 95°C for 3 min, 95°C for 30 s, 60°C for 45 s, and 72°C for 10 min. The LoopSeq-adapted cDNA was then purified using 0.6× SPRI and resuspended in 20 μL of pre-warmed Hybridization Mix. The bead slurry was then enriched by a human core exome capture procedure (Twist Bioscience, CA). In brief, 5 μL of Buffer EB, 5 μL blocker solution, 6 μL LoopSeq adapter blocker, 4 μL biotinylated exome probe solution, and 30 μL hybridization enhancer were added to the bead slurry and incubated at 95°C for 5 min followed by 70°C for 16 hr. The hybridized cDNA was then captured by streptavidin beads following the protocol recommended by the manufacturer. Then, 10 μL of the probe captured, LoopSeq-adapted cDNA was then combined with 5 μL of barcode oligo primers, and 35 μL of a PCR Barcoding cocktail for amplification using the following parameters: 95°C for 3 min, 12 cycles of 95°C for 30 s, 68°C for 45 s, 72°C for 2 min. The amplified captured cDNA underwent a 0.6× SPRI purification and was eluted in 30 μL of Buffer EB. This product is diluted with Buffer EB to adjust for a desired long-read barcode complexity. Then, 2 μL of each diluted product were independently combined with 18 μL of an Amp Mix S and an Amplification Additive Master mix and underwent thermocycling using the following parameters: with a 100°C heated lid, amplify samples at 95°C for 3 min, followed by 22 cycles of 95°C for 30 s, 60°C for 45 s, and 72°C for 2 min. Then, 10 μL of each amplification reaction was pooled and underwent a 0.6× SPRI purification before elution in 40 μL of Buffer EB.

## LOOP UMI distribution and library construction

Then, 30 µL of the eluate was combined with 10 µL distribution mix and 4 µL distribution enzyme and incubated at 20°C for 15 min. The reaction was then terminated by heating to 75°C for 5 min. The distributed UMIs were activated by incubating the reaction with 56 µL of activation mixture cocktail at 20°C for 2 hr and neutralized with the addition of 6 µL of neutralization enzyme and heating at 37°C for 15 min. The samples were then 0.8× SPRI purified to remove small undistributed UMI DNA. Then, 35 µL of the LOOP UMI-distributed cDNA was fragmented with 15 µL of fragmentation enzyme master mix at 32°C for 5 min, followed by 65°C for 30 min. The fragmented LOOP UMI-distributed cDNA was ligated with 40 µL of Ligation master and 10 µL of Ligation Enzyme at 20°C for 15 min. The ligated DNA was 0.6× SPRI purified, eluted in 20 µL of Buffer EB, and amplified using 25 µL of Index Master Mix and 5 µL of index primers in the following condition: 95°C for 3 min, then 12 cycles of 95°C for 30 s, 65°C for 45 s, and 72°C for 30 s. The amplified product undergoes a final 0.6× SPRI purification and 20 µL elution in Buffer EB. After the final short-read library was quantified via qPCR and assessed for quality using a Agilent bioanalyzer 2100, the library cocktail was sequenced on an Illumina NovaSeq.

## TaqMan qRT-PCR assay for fusion genes

Total RNA was extracted using TRIzol (Invitrogen, CA; *Chen et al., 2015*; *Yu et al., 2019a*; *Yu et al., 2019b*; *Yu et al., 2014b*; *Zuo et al., 2017*). Then, 2 µg of RNA were used to synthesize the first-strand cDNA with random hexamer primers and Superscript II (Invitrogen). Also, 1 µL of each cDNA sample was used for TaqMan PCR (Eppendorf RealPlex Mastercycler and Applied Biosystems QuantStudio 3) with 50 heating cycles at 94°C for 30 s, 61°C for 30 s, and 72°C for 30 s using the following primer sequences: GAGTGATATCAGACACCGAGC/TTTCTGGGACTCCCTAGACCA and the following TaqMan probe: 5'-/56-FAM/AA GCTCTCT/ZEN/CCAACGGTTGGA/3IABkFQ/–3' for PDCD6-CCDC127, AGGAAGGTGGTGGTGTGCGA/TTGGGTGAAACTCCACAGCCA and the following TaqMan probe: 5'-/56-FAM/AACGGCACC/ZEN/GGGACAACAAAT/3IABkFQ/–3' for ACTR2-EML4, CCACAGGAAAGAAGTGTCAGTC/GTTATGGAGTTTTCAACATGGGG and the following TaqMan probe: 5'-/56-FAM/AAGCTCTCT/ZEN/CCAACGGTTGGA/3IABkFQ/–3' for PLG-FGG in an Eppendorf RealPlex thermocycler.

## De novo assembly of long-read transcripts, short-read trimming, and long-read alignment

The long-read transcripts were assembled using SPADES (*Bankevich et al., 2012*) from a Python script with the following parameters: command = spades.py -k 21,33,55,77,99,127 -t 1 `--careful --sc --pe1-1` left.fq `--pe1-2` right.fq `--pe1-s` unpaired.fq -o spades_output `--disable-gzip-output`. Short-read trimming was performed using Trimmomatics (*Bolger et al., 2014*) with the following parameters: command = ['java -jar '+pipeline.prog_path + '/Trimmomatic-0.36/trimmomatic-0.36.jar PE -threads 32 -trimlog '+trim_log_file + ' ','../'+pipeline.input_params['raw_file_R1'], '../'+pipeline.input_params['raw_file_R2'],' '.join(trim_output_files), 'ILLUMINACLIP:'+pipeline.prog_path + '/JAStrim.fa:2:40:14:3:true TRAILING:20 SLIDINGWINDOW:4:15 MINLEN:36']. Long-read alignment through BLAST was performed from a Python script with default parameters: blastn -db<reference database> -query contig_list.fa -perc_identity=<pct_id_threshold> -qcov_hsp_perc = <qcov_threshold> -max_target_seqs = <max_seqs> -num_threads=16 -outfmt=6 > mapping.blst.

## Gene and isoform expression analysis on LoopSeq single-cell data

Paired HCC samples (HCC and benign liver) were compared by LoopSeq single-cell transcriptome sequencing. In total, six runs were performed on each library and were pooled together for analysis. LoopSeq long-reads were analyzed using SQANTI (*Tardaguila et al., 2018*) for gene and isoform annotation (human reference hg38). Based on the cell (10×) barcodes and molecule (Loop) barcodes from both long-reads and short-reads, long-read molecules were able to be assigned to cells and unique molecules. UMIs were quantified at both gene and isoform levels based on the SQANTI annotation and cell/molecule assignment. Valid cells were defined as cells with more than 1000 long-read molecules. In total, 162 normal cells and 285 tumor cells were used for the downstream analysis, with the highest number of long-read transcripts reaching 56,745/cell for HCC and 49,476/cell for benign liver.

Single-cell expression data were integrated using the R/Bioconductor package *Seurat* (*Hao et al., 2021*). Expression SDs per gene and isoforms across all the cells were calculated. Genes or isoforms (SDGs and SDIs, respectively) with SDs above a certain threshold were defined. Cell clustering was performed based on the expression profile of these selected genes/isoforms and was visualized using UMAP algorithm (*McInnes et al., 2018*). Markers identifying each cluster were detected by comparing the cells in a specific cluster and all the other cells not in that cluster.

## Mutation calling on whole-exome sequencing data

Whole-exome sequencing (WES) was performed on the same HCC patient with three libraries: HCC cells, benign liver cells, and normal gallbladder tissue. For each library, low-quality reads and adapter sequences were trimmed from the raw sequencing reads using the tool Trimmomatic (*Bolger et al., 2014*). After pre-processing, the surviving reads were aligned to human reference genome hg38 using Burrows-Wheeler Aligner. Picard (http://broadinstitute.github.io/picard; *Broad Institute, 2019*) Version: 2.18.12; (RRID:SCR_006525) was employed to sort the aligned files and mark duplicates. Alternative (single-nucleotide variants or SNPs) calling was then performed using the SAMtools *mpileup* function (*Li, 2011*; *Li et al., 2009*), and somatic mutations on paired samples (normal gallbladder vs. benign liver pair, or normal gallbladder vs. HCC liver pair) were called using the GATK *MuTect2* function (*McKenna et al., 2010*). Amino acids were annotated to those alternatives using SnfEff (*Cingolani et al., 2012*). The mutations of interest were selected by the following criteria: (1) the mutation must be non-synonymous or stop gain; and (2) the mutation must be present in either HCC or benign liver samples but not in the normal gallbladder tissue. These mutations will serve as a validation set for the long-read single-cell transcriptome data analysis. All the pipelines were run by default parameter settings.

## Mutation isoform analysis on LoopSeq single-cell data

Single-cell transcriptome long-reads were aligned to human reference genome hg38 by long-read aligner Minimap2 (*Li, 2018*). Alternative (single-nucleotide variants or SNPs) calling was then performed using the SAMtools *mpileup* function (*Li, 2011*; *Li et al., 2009*). To avoid sequencing errors, RNA-editing events, and non-tumor-specific mutations, only mutations validated by the whole-exome sequencing method were used. Based on the long-read cell barcode, the number of reference reads and alternative reads per mutation position and per valid single cell were quantified for the downstream analysis. The mutation rate was calculated by the number of alternative reads over the total reads (sum of reference and alternative reads). Based on the SQANTI annotation (*Tardaguila et al., 2018*) of the long-read, mutations were quantified both at gene and isoform levels.

The SD of the mutation rate (per gene or isoform mutation) was calculated across all the valid cells. High variable mutations were defined as those with SDs > 0.4. These SD mutations were then used as features to perform cell clustering and UMAP visualization. Isoform-level mutation analysis resolved three clusters based on SD ≥ 0.4 mutations. Mutations per cluster were defined by the mutations that exist in at least five cells of that cluster, but not in any of the cells in the other two clusters. A total of 104 mutations were found among the three clusters. Based on these mutations, additional clustering was performed, and eight sparse clusters were detected and used to group the cells. HLA-related mutations were specifically examined and quantified across the eight clusters. Evolution flowcharts were generated based on the progression of the mutation sites.

## Fusion transcript detection on LoopSeq single-cell data

Fusion transcripts were called by two pipelines: (1) SQANTI (*Tardaguila et al., 2018*) performs the fusion annotation on the long-read sequencing data. (2) Based on the Minimap2 (*Li, 2018*) alignment and hg38 UCSC annotation file, fusions were called from the long-reads that were aligned to two genes. Based on all the fusion calling, the following filtering criteria were applied: (1) eliminate the fusions where the head and tail genes were in *cis*-direction and were less than 40 kb apart; (2) eliminate the fusions whose head genes have more than two tail partners in all the fusion callings; (3) eliminate the fusions whose tail genes have more than two head gene partners in all the fusion callings; (4) only keep those fusions whose joining points are located at the edge of the exons; and (5) fusions must be detected in at least two cells. These selected fusions and experimentally validated fusions were subsequently used for downstream analysis.

## Integrative analysis to combine expression, mutation, and fusion data

High SD expression genes (or isoforms), high SD mutation genes (or isoforms), and selected fusion transcripts were integrated. UMAP (*McInnes et al., 2018*) cell visualization was applied, combining all three feature sets to perform the cell clustering. Data visualization was performed using the R/Bioconductor package *ComplexHeatmap* (*Gu et al., 2016*) and *ggplot2* (*Wickham, 2016*).

## Bioinformatic pipeline script

```
OUTPUT=$outPath/alignment.arg.sorted.dedup.realigned.fixmate.bam SO
=coordinate CREATE_INDEX =true
```

# base quality score recalibration

```
knownSite1=$refPath/GATKreference/hg38/dbsnp_138.hg38.vcf

knownSite2=$refPath/GATKreference/hg38/resources_broad_hg38_v0_1000G_
phase1.snps.high_confidence.hg38.vcf

knownSite3=$refPath/GATKreference/hg38/Mills_and_1000G_gold_standard.
indels.hg38.vcf

java -Xmx20g -Djava.io.tmpdir=$tmpFolder -jar $GATK -T
BaseRecalibrator -R $refFile -knownSites $knownSite1 -knownSites
$knownSite2 -knownSites $knownSite3 -I $outPath/alignment.arg.sorted.dedup.
realigned.fixmate.bam -o
$outPath/recal_data.table -cov ReadGroupCovariate -cov
QualityScoreCovariate -cov CycleCovariate
```

# print reads

```
java -Xmx20g -Djava.io.tmpdir=$tmpFolder -jar $GATK -T PrintReads -R
$refFile -BQSR $outPath/recal_data.table -I
$outPath/alignment.arg.sorted.dedup.realigned.fixmate.bam -o
$outPath/alignment.arg.sorted.dedup.realigned.fixmate.recal.bam
```

## MuTect2 for somatic mutation calling

```
dbSNPfile=$refPath/GATKreference/hg38/dbsnp_138.hg38.vcf

COSMICfile=$refPath/MuteckRef/GRCh38/CosmicCodingMuts_chr.vcf

java -Xmx20g -Djava.io.tmpdir=$tmpFolder -jar $GATK -T MuTect2 -nct 2-R
$refFile -I:tumor $outPathT/alignment.arg.sorted.dedup.realigned.fixmate.
recal.bam -
I:normal
$outPathN/alignment.arg.sorted.dedup.realigned.fixmate.recal.bam --cosmic
$COSMICfile --dbsnp
$dbSNPfile -o $outPath/NormalTumorPair.muTect2.vcf
```

## snpEff and snpSift for SNP and AA annotation
```
java -Xmx40g -jar /zfs2/sliu/tools/snpEff/snpEff.jar GRCh38.86 $sample.vcf
> $sample.anno.vcf
```

## Acknowledgements

We thank Songyang Zheng for technical support. This work was in part supported by grants from the National Cancer institute (1R56CA229262-01 to JHL), National Institute of Digestive Diseases and Kidney (P30- DK120531-01), National Institute of Health (UL1TR001857 and S10OD028483), and The University of Pittsburgh Clinical and Translational Science Institute.

## Additional information

### Competing interests

Tuval Ben-Yehezkel: He is an employee of Element Biosciences, Inc. Caroline Obert: She is an employee of Element Biosciences, Inc. Mat Smith: MS is an employee of Element Biosciences, Inc. The other authors declare that no competing interests exist.

### Funding

| Funder | Grant reference number | Author |
|---|---|---|
| National Cancer Institute | 1R56CA229262-01 | Jian-Hua Luo |
| National Institute of Diabetes and Digestive and Kidney Diseases | P30- DK120531-01 | Jian-Hua Luo |
| National Institutes of Health | UL1TR001857 and S10OD028483 | Silvia Liu |
| University of Pittsburgh | | Jian-Hua Luo |

The funders had no role in study design, data collection and interpretation, or the decision to submit the work for publication.

### Author contributions

Silvia Liu, Software, Formal analysis, Funding acquisition; Yan-Ping Yu, Supervision, Investigation; Bao-Guo Ren, Formal analysis, Methodology; Tuval Ben-Yehezkel, Software, Investigation; Caroline Obert, Investigation, Methodology; Mat Smith, Wenjia Wang, Methodology; Alina Ostrowska, Alejandro Soto-Gutierrez, Resources; Jian-Hua Luo, Conceptualization, Supervision, Investigation, Methodology, Writing – original draft, Writing – review and editing

### Author ORCIDs

Jian-Hua Luo ⓘ http://orcid.org/0000-0002-3189-4225

### Ethics

Human subjects: The hepatocellular carcinoma and benign liver samples were obtained from the Pittsburgh Liver Research Center biospecimen core of University of Pittsburgh in compliance with institutional regulatory guidelines. The informed consent exemptions and protocol have been approved by the Institution Review Board of University of Pittsburgh. (Study#19030097).

Reviewer #1 (Public Review): https://doi.org/10.7554/eLife.87607.3.sa1
Reviewer #2 (Public Review): https://doi.org/10.7554/eLife.87607.3.sa2
Author Response https://doi.org/10.7554/eLife.87607.3.sa3

## Additional files

### Supplementary files

- Supplementary file 1. Mutation gene expressions.
- Supplementary file 2. Mutation isoform expressions.
- Supplementary file 3. Mutation isoform expressions in ≥ 5 transcriptomes.
- Supplementary file 4. Genes with SD ≥ 0.5.

- Supplementary file 5. Genes with SD ≥ 0.8.
- Supplementary file 6. Genes with SD ≥ 1.0.
- Supplementary file 7. Genes with SD ≥ 1.4.
- Supplementary file 8. Isoforms with SD ≥ 0.5.
- Supplementary file 9. Isoforms with SD ≥ 0.8.
- Supplementary file 10. Isoforms with SD ≥ 1.0.
- Supplementary file 11. Isoforms with SD ≥ 1.4.
- Supplementary file 12. Gene expressions with SD ≥ 1.0, mutation gene expressions with SD ≥ 0.4 and 19 fusion genes.
- Supplementary file 13. Signaling pathways impacted by differential gene expressions.
- MDAR checklist

## Data availability

LOOP Single-cell transcriptome sequencing data has been deposited to Gene Expression Omnibus (GEO) database with access ID: GSE223743.

The following dataset was generated:

| Author(s) | Year | Dataset title | Dataset URL | Database and Identifier |
|---|---|---|---|---|
| Liu S, Yu Y, Ren B, Yehezkel TB, Colbert C, Wang W, Ostrowska A, Soto-Gutierrez A, Luo J | 2023 | Ultra-low error synthetic long-read single-cell sequencing reveals expressions of hypermutation clusters of isoforms in human liver cancer cells | https://www.ncbi.nlm.nih.gov/geo/query/acc.cgi?acc=GSE223743 | NCBI Gene Expression Omnibus, GSE223743 |

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
