## [Editor Report · eLife assessment]

The authors pair single-cell sequencing technology with the LoopSeq synthetic long-read method to examine samples of hepatocellular carcinoma and benign liver, with the goal of identifying mutations and fusion transcripts specific to cancer cells. The authors present a **valuable** resource, and the overall support for the major claims is **solid**.

---

## [Referee Report · Reviewer #1 (Public Review)]

In the manuscript "Long‐read single‐cell sequencing reveals expressions of hypermutation clusters of isoforms in human liver cancer cells", S. Liu et al present a protocol combining 10x Genomics single-cell assay with Element LoopSeq synthetic long-read sequencing to study single nucleotide variants (SNVs) and gene fusions in Hepatocellular carcinoma (HCC) at single‐cell level. The authors were the first to combine LoopSeq synthetic long‐read sequencing technology and 10x Genomics barcoding for single cell sequencing. For each cell and each somatic mutation, they obtain fractions of mutated transcripts per gene and per each transcript isoform. The manuscript states that these values (as well as gene fusion information) provide better features for tumor-normal classification than gene expression levels. The authors identified many SNVs in genes of the human major histocompatibility complex (HLA) with up to 25 SNVs in the same molecule of HLA‐DQB1 transcript. The analysis shows that most mutations occur in HLA genes and suggests evolution pathways that led to these hypermutation clusters.

---

## [Referee Report · Reviewer #2 (Public Review)]

In the present study, Liu et al present an analysis of benign and HCC liver samples which were subjected to a new technology (LOOP-Seq) and paired WES. By integrating these data, the authors find isoforms, fusions and mutations which uniquely cluster within HCC samples, such as in the HLA locus, which serve as candidate leads for further investigation. The main appeal of the study is in the potential of LOOP-Seq as a method to present isoform-resolved data without actually performing long-read sequencing.

Comments on revised version:

I made several comments on the previous version which have been adequately addressed.

---

## [Author Response]

The following is the authors’ response to the original reviews.

We thank the reviewers and editors for their constructive comments on the manuscript. We have extensively revised the manuscript based on these concerns and comments. The followings are the specific answers.

**Public Reviews:**

**Reviewer #1 (Public Review):**
In the manuscript "Long‐read single‐cell sequencing reveals expressions of hypermutation clusters of isoforms in human liver cancer cells", S. Liu et al present a protocol combining 10x Genomics single-cell assay with Element LoopSeq synthetic long-read sequencing to study single nucleotide variants (SNVs) and gene fusions in Hepatocellular carcinoma (HCC) at single‐cell level. The authors were the first to combine LoopSeq synthetic long‐read sequencing technology and 10x Genomics barcoding for single cell sequencing. For each cell and each somatic mutation, they obtain fractions of mutated transcripts per gene and per each transcript isoform. The manuscript states that these values (as well as gene fusion information) provide better features for tumor-normal classification than gene expression levels. The authors identified many SNVs in genes of the human major histocompatibility complex (HLA) with up to 25 SNVs in the same molecule of HLA‐DQB1 transcript. The analysis shows that most mutations occur in HLA genes and suggests evolution pathways that led to these hypermutation clusters. Yet, very little is said about novel isoforms and alternative splicing in HCC cells, differences in isoform ratio between cells carrying different mutations, or diversity of alternative isoforms across cells. While the manuscript by Liu et al. presents a promising combination of technologies, it lacks significant insights, a comprehensive introduction, and has significant problems with data description and presentation.

Answer: Thanks for the precious suggestion. Our long-read single-cell sequencing has discovered an average of 442 novel isoform transcripts per benign liver cell and 450 novel isoform transcripts per HCC cell per SCANTI v1.2 analysis. These are stated in the revised manuscript. The alternative splicing was detected by differential isoform expression as demonstrated in supplemental figures 6 and 7 and supplemental tables 8-11. The examples of differences in isoform ratio between cells carrying different mutations are now shown by DOCK8 and STEAP4 (figure 5 in the revised manuscript). A new section was added in the results to discuss the mutation expression of these two genes. The diversity of isoforms of the selected genes is shown in Supplemental Figure 10.

This study showed how mutations in the same allele evolved in liver cancer. In particular, HLA hypermutations were found to develop from some specific sites of the molecules into large clusters of mutations in the same molecules. A new paragraph of introduction was added about the role of mutations in human cancer development. We also revised the figures to present the information better. All the HLA genes expressed only one known isoform, as shown in Figure 4 and Supplemental Figure 3, regardless of mutations.

Major comments:1. The introduction section is scarce. It lacks description of important previous works focused on clustered mutations in cancers (for example, PMID35140399), on deriving the process of cancer development through somatic evolution (PMID32025013, from single cell data PMID32807900). Moreover, some key concepts e.g. mutational gene expression and mutational isoform expression are not defined. The introduction and the abstract contain slang expressions e.g. "protein mutation', a combination of terms I teach my students not to use.

Answer: We appreciate the reviewer for the idea of more solid background introduction and term definition. We added a new paragraph in the introduction section to introduce the role of mutations andhypermutations in human cancers. Some important work has been cited. We added a new section in the "Methods" to define "mutation gene expression share" and "mutation isoform expression share". "Protein mutation" has been replaced by "genetic mutation".

1. In the results section, to select the mutations of interest, the authors apply UMAP dimensionality reduction to the mutation isoforms expression and cluster samples in UMAP space, then select the mutations that are present only in one cluster, then apply UMAP to the selected mutations only and cluster the samples again. The motivation for such a procedure seems unclear, could it be replaced with a more straightforward feature selection?

Answer: Thanks for raising up this important question. The goal of the analysis is an unbiased classification of the cell populations in the samples. We found that by removal of mutated isoform expressions that were at similar levels of all cells, the UMAP clustering generated clear segregation of three population cells. When the unique mutated isoform expressions from each group were applied, it generated highly distinct 8 groups of cells, with each group having a distinct mutation isoform expression pattern. If we force known knowledge into the mix of the analysis, it may generate unwanted bias. Specifically, the first UMAP was performed in an unbiased way to cluster cells, while the second step is a supervised approach by selecting the unique mutations in each cluster to identify the classifiers. The second UMAP matches the Benign/HCC labeling well.

1. As I understand, the first "mutated isoform"-based UMAP clustering was built from expression levels of 205 "mutational isoforms". What was the purpose and outcome of the second "mutated isoform"based UMAP clustering (Figure 2E)? In the manuscript the authors just describe the clusters and do not draw any conclusions or use the results of the clustering anywhere further.

Answer: Thanks for pointing this out. Figure 2E was generated from unique mutation isoform expressions in groups A, B, and C from Figure 2D. The purpose of Figure 2E is to investigate whether these unique mutation isoforms can further classify the cell populations free of prior biological knowledge. We added a sentence in the revision to clarify the purpose of the clustering. The conclusion from this analysis, including Figure 2F and Figure 3 (which is an extension of Figure 2E), is that HLA mutation isoform expressions dominated the classifications of cell populations.

1. The authors just cluster the data three times based on expression levels of different sets of "mutational isoforms" and describe the clusters. What do we need to gather from these clustering attempts besides the set of 113 mutations used for further analysis? What was the point of the reclusterings? Did the authors observe improvement of the classification at each step?

Answer: Thanks for asking this important question. The improvement of re-clustering to classify cell populations is the obvious segregation of 8 different groups of cells without any manual classification through prior knowledge. The distances among groups were far apart in comparison to the first clustering (figure 2B). Detailed subclassifications were achieved on cell populations that otherwise could not be segregated based on the first clustering.

1. The alignment of short reads generated from hypermutated transcriptomes is non-trivial. The proposed approach could address the issue without the need for whole genome sequencing and offer insights about the cancer development through somatic evolution. Why didn't the authors use modern phylogenetic approaches in the "Evolution of mutations in HLA molecules" section or at least utilize the already performed clustering to infer cell lineages?

Answer: We appreciate for the great question. For a single molecule mutation evolution, single gene clustering may not produce a desirable and robust effect. A simple evolution snowball chart in Figure 4B may be easier to be understood.

1. I am not sure I understood the definition of "mutated gene expression levels" and "mutated isoform expression levels" in the "Mutational gene expression and fusion transcript enhanced transcriptome clustering of benign hepatocytes and HCC" section. The authors mention that gene lists included all the isoforms within the same range of standard deviation. If I understand it correctly, they are equal if there is only one expressed transcript isoform. In that case, this overlap is not surprising at all.

Answer: We thank the reviewer for the great question. The definition of mutation gene expression level, mutation isoform expression level, and fusion gene expression level are now defined in the "Methods" section. In all HLA mutation transcripts, there were multiple transcripts with or without mutations for a single dominant isoform.

1. "To investigate the roles of gene expression alterations that were not accompanied with isoform expression changes, UMAP analyses were performed based on the non‐overlapped genes." Venn diagrams (Sup Figure 8) show that there are much less "non-overlapped genes" than "genes that showed both gene and isoform level changes" for each SD threshold (for example, for SD>=0.8 59 vs 275). Could that be the reason why clustering based on the former group is worse i.e the cancer and normal cells are separated less clearly?

Answer: The number of (attributes) genes could be a contributing factor in the segregation of cell populations. However, the number of attributes is not the underlying reason for worse performance for gene only classifier because much smaller isoforms/genes (22) overlap in SD>=1 outperformed a large number of genes (59) with SD>=0.8. It suggested that 59 gene expression classifier is less efficient in segregating the cell populations. To address this concern, we took SD>=0.8 as an example for demonstration if we subsampled the 275 overlapped genes/isoforms to 59 (equal to 59 non-overlapped genes in terms of number), we can still get better separation than the 59 DEG only. We repeated this subsampling process for three times. Similar results were found. The new data were inserted into supplemental Figure 8

**Reviewer #2 (Public Review):**
In the present study, Liu et al present an analysis of benign and HCC liver samples which were subjected to a new technology (LOOP-Seq) and paired WES. By integrating these data, the authors find isoforms, fusions and mutations which uniquely cluster within HCC samples, such as in the HLA locus, which serve as candidate leads for further investigation. The main appeal of the study is in the potential of LOOPSeq as a method to present isoform-resolved data without actually performing long-read sequencing. While this presents an exciting new method, the current study lacks systematic comparisons with other technologies/data to test the robustness, reproducibility and utility of LOOPSeq. Further, this study could be further improved by giving more physiologic context and examples from the analyses, thus providing a new resource to the HCC community. A few suggestions based on these are below:

Answer: We appreciate the reviewer to raise up all the important questions and the great suggestions. TheLOOPseq technology was compared with Oxford nanopore and PacBio long-read sequencing in our previous study. We have cited analysis in the introduction section of the paper. HLA mutation clusters in the single molecules are our finding with major physiological significance since these mutations may help liver cancer cells evade immune surveillance. We have extensively discussed the potential impact of these mutations on cancer development in the discussion. In addition, we added a new section of DOCK8 and STEAP4 mutation expressions in the results (page 11, new Figure 5) that are highly relevant to the pathogenesis of HCC.

1. A primary consideration is that this seems to be the first implementation of LOOP-Seq, where the technology, while intriguing, has not been evaluated systematically. It seems like a standard 10x workflow is performed, where exons are selectively pulled down and amplified. Subsequent ultra-deep sequencing is assumed to give isoform-resolution of the sc-seq data. To demonstrate the utility of the approach it would benefit the study to compare the isoform-resolved results with studies where long-read sequencing was actually performed(ex: https://journals.lww.com/hep/Fulltext/2019/09000/Long_Read_RNA_Sequencing_Identifies_Alternativ e.19.aspx, https://www.jhep-reports.eu/article/S2589-5559(22)00021-0/fulltext, https://journals.plos.org/plosgenetics/article?id=10.1371/journal.pgen.1010342). Presumably, a fair amount of overlap should occur to justify the usage.

Answer: We have discussed the utility of the methodology in comparison with the previous studies by these three groups in the revision (results, page 12).

1. Related to this point, the sc-seq cell types and benign vs HCC genes should be compared with the wealth of data available for HCC sc-seq (https://www.nature.com/articles/s41467-022-322833, https://www.nature.com/articles/s41598-021-84693-w). These seem to be important to benchmark the technology in order to demonstrate that the probe-based selection and subsequent amplification does not bias cell type definition and clustering. In particular, https://www.nature.com/articles/s41586021-03974-6 seems quite relevant to compare mutational landscapes from the data.

Answer: This is a great point. The consistency probe-based analysis was demonstrated in our previous analyses and the analyses mentioned in the comments. We further discussed it in the results section of the paper (page 12).

1. From the initial UMAP clustering, it will be important to know what the identities are of the cells themselves. Presumably, there is quite a bit of immune cells and hepatocytes, but without giving identities, downstream mechanistic interpretation is difficult.

Answer: When mutation analyses were combined with cell marker analysis, i.e., immune marker positive but negative in HLA mutation, we found only one bona fide immune cell in the HCC sample. Thus, immune cells may not be significant in the current analysis.

1. In general, there are a fair amount of broad analyses, such as comparisons of hierarchical clustering of cell types, but very little physiologic interpretations of what these results mean. For example, among the cell clusters from Fig 6, knowing the pathways and cell annotations would help to contextualize these results. Without more biologically-meaningful aspects to highlight, most of the current appeal for the manuscript is dependent on the robustness of LOOP-seq and its implementation.

Answer: To address this comment, a new pathway analysis was performed on the cluster results of Figure 6. A new supplemental table was generated. The results are now discussed on page 13.

1. Many of the specific analyses are difficult and the methods are brief. Especially given that this technology is new and the dataset potentially useful, I would strongly recommend the authors set up a git repository, galaxy notebook or similar to maximize utility and reproducibility

Answer: The script file has been uploaded to GIT to facilitate the reproducibility of the analysis. We also added a new pipeline description script in the methods (pages 19-20).

1. The authors claim that clustering between benign and HCC samples was improved by including isoform & gene (Suppl fig 8). This seems like an important conclusion if true, especially to justify the use of longread implementation. Given that the combination of isoform + gene presents ~double the number of variables on which to cluster, it would be important to show that the improved separation on UMAP distance is actually due to the isoforms themselves and not just sampling more variables from either gene or isoform

Answer: The number of (attributes) genes could be a contributing factor in the segregation of cell populations. However, the number of attributes is not the underlying reason for worse performance for gene only classifier because much smaller isoforms/genes (22) overlap in SD>=1 outperformed a large number of genes (58) with SD>=0.8. It suggested that 58 gene expression classifier is less efficient in segregating the cell populations. To address this comment, we performed random subsampling to reduce the isoform/gene overlap iterates, similar results were obtained. A new supplemental figure was generated to reflect the new analyses.

1. SQANTI implementation to identify fusions relevant for the HCC/benign comparison. How do the fusions compare with those already identified for HCC? These analyses can be quite messy when performed on WES alone so it seems that having such deep RNA-seq would improve the capacity to see which fused genes are strongly expressed/suppressed. This doesn't seem as evident from current analysis. There are quite a bit of WES datasets which could be compared: https://www.nature.com/articles/ng.3252, https://www.nature.com/articles/s41467-01803276-y

Answer: Exome sequencing is not an ideal tool to identify fusion genes. Very few fusion genes have been discovered based on RNA sequencing so far. The fusion genes discovered in the study appeared mostly novel. No exome sequencing was involved in the identification of fusion genes.

1. Figure 4 is fairly unclear. The matrix graphs showing gene position mutations are tough to interpret and make out. Usually, gene track views with bars or lollipop graphs can make these results more readily interpretable. Also, how Figure 4 B infers causal directions from mutations is unclear.

Answer: We appreciate the reviewer for pointing this out. We have revised the diagram in Figure 4A to reflect the proper distance between the mutations in HLA-DQB1 NM_002123. Since these are the positions in the same alleles (protein), the gene track view or lollipop graph may not show that properly.The mutation clusters started from an isolated mutation, and mutation did not revert to wild type sequence after occurring. Based on these two principles, we showed several mutation accumulation pathways leading to hypermutation clusters.

**Reviewer #3 (Public Review):**
The Liu, et al. manuscript focuses on the interesting topic of evaluating in an almost genome-wide-scale, the number of transcriptional isoforms and fusion gene are present in single cells across the annotated protein coding genome. They also seek to determine the occurrences of single nucleotide variations/mutations (SNV) in the same isoform molecule emanating from the same gene expressed in normal and normal and hepatocellular carcinoma (HCC) cells. This study has been accomplished using modified LoopSeq long‐read technology (developed by several of the authors) and single cell isolation (10X) technologies. While this effort addresses a timely and important biological question, the reader encounters several issues in their report that are problematic.:1. Much of the analysis of the evolution of mutations results and the biological effects of the fusion genes is conjecture and is not supported by empirical data. While their conclusions leave the reader with a sense that the results obtained from the LoopSeq has substantive biological implications. However, they are extended interpretations of the data. For example: The fusion protein likely functions as a decoy interference protein that negatively impacts the microtubule organization activity of EML4.(pg 9)... and other statements presented in a similar fashion.

Answer: We thank the reviewer for the helpful comment. The mutation results were experimentally validated by exome sequencing on the same samples. Furthermore, these mutations were filtered by requiring their presence in three different transcriptomes. The biological significance of these mutations is probably the subject of investigation in the next phase. Since a large number of HLA mutations did not occur overnight, the analysis of the accumulation pathways for these mutations was warranted, given the extensive evidence of such a process. The impact of mutations on HLA molecules appeared obvious and should be discussed. For ACTR2-EML4 fusion, we revised it as "The loss of microtubule binding domain may negatively impact the microtubule organization activity of EML4 domain of the fusion protein." We only discussed the obvious impact due to the loss of a large protein domain.

2, LoopSeq has the advantage of using short read sequencing analyses to characterize the exome capture results and thus benefits from low error rate compared to standard long-read sequencing techniques. However, there is no evidence obtained from standard long read sequencing that the isoforms observed with LoopSeq are obtained with parallel technologies such as long read technologies. It is not made clear how much discordance there is in comparing the LoopSeq results are with either PacBio or ONT long read technologies.

Answer: The comparative analyses among LOOPSeq, Oxford nanopore, and PacBio sequencing were performed in our previous study. We have cited the study in our introduction.

1. There is no proteome evidence (empirically derived or present in proteome databases) from the HCC and normal samples that confirms the presence or importance of the identified novel isoforms, nor is there support that indicate that changes in levels HLA genes translate to effects observed at the protein level. Since the stability and transport differences of isoforms from the same gene are often regulated at the post-transcriptional level, the biological importance of the isoform variations is unclear.

Answer: Given the transcriptome sequencing data, we can only focus on the isoform variation analysis but not directly link to the protein level variation because of the post-transcriptional level regulation. We discussed this in the revised manuscript (page 14).

4 It is unclear why certain thresholds were chosen for standard deviation (SD) <0.4 (page 5), SD >1.0 (pg 11).

Answer: The threshold is flexible and arbitrary. We showed different thresholds, and the same conclusion holds. We just choose the thresholds with better separation and a reasonable number of genes/isoforms for the downstream analysis. (Supplemental Figure 6-7 with different thresholds and supplemental tables 4-12).

1. HLA is known to accumulate considerable somatic variation. Of the many non-immunological genes determined to have multiple isoforms what are the isoform specific mutation rates in the same isoform molecule? Are the HLA genes unique in the number of mutations occurring in the same isoform?

Answer: We thank the reviewer for this important suggestion. We now show mutation expression patterns in isoforms of DOCK8 and STEAP4 in Figure 5. A new section is added to discuss the mutation expression of these two genes. As shown in supplemental figure 10, HLA-DQB1, HLA-DRB1, HLA-B, and HLA-C, have only one known isoform detected,

**Editorial comments:**
The present study pairs single-cell seq with LoopSeq synthetic long-read sequencing on samples of HCC and benign liver to identify mutations and fusion transcripts specific to cancer cells. The authors present a potentially important resource; however the overall support remains incomplete.While the approach of evaluating isoform-specific changes at the cellular level to cancer seeks to address a timely and important topic, there is currently incomplete evidence in support of the major claims in the manuscript. In particular, major recommendations to provide stronger support for the combination of technologies and interpretation regarding cancer-associated genomic changes include: (1) systematic evaluation of UMAP-based clustering methods, to what subsets of data they are applied and subsequent interpretations, (2) direct comparisons of results with additional methods to quantify long-read sequencing data and those evaluating mutational consequences of HCC progression and (3) detailed expansion of the description of methods and rationale for selecting specific parameters and cell types for further analyses. Including these changes would significantly strengthen the support for utility of combining 10x single-cell with Loop-seq and provide compelling evidence for usage of this resource in dissecting HCC-associated molecular changes.

Answer: We appreciate the frank and constructive comments. The goal of UMAP is to obtain biological knowledge through unbiased data selection. Systematically, we select classifiers without any prior knowledge (blind to the samples). In our case, classifiers with high standard deviation across all the cells were chosen. We stressed this in the result section. The comparison among LOOPSeq, PacBio, and Oxford nanopore was made in our previous study. We cited that analysis in this paper. Analysis detail and pipelines were added in the revised manuscript to improve the reproducibility. The mutation expression analysis was quite clear-cut. The clustering classified the HCC and benign liver cells by itself and identified a few cancer cells in the benign liver sample. All these were accomplished without applying any knowledge.

**Reviewer #1 (Recommendations For The Authors):**
Overall, there are numerous problems with data presentation and insufficient description, which authors could fix.1. Figure 4. A. It would be more clear if the figure showed the distribution of mutations in the molecule. Otherwise, it's hard to see if we see clusters of mutations or just 25 mutations spread uniformly across the transcript. B. It's unclear what the reader needs to take away from these columns of numbers.

Answer: The mutation positions are now presented as proportion to the location in a molecule. Column B is the distribution of mutation molecules from left panel in each cluster of cells (from Figure 3A) and their sample origin (HCC or benign liver). We clarify it a little more in the legend of Figure 4A.

1. As a reader, I did not understand how "mutated gene expression levels" and "mutated isoform expression levels" were calculated in terms of sequenced long reads

Answer: We defined the term and calculations in the methods section of the revised manuscript.

1. Page 6 "genes involving antigen presentation"

Answer: The full sentence of the subtitle is" Mutations of genes involving antigen presentation dominated the mutation expression landscape."

1. Page 6 "These unique mutational isoforms" - how are these isoforms unique?

Answer: We take away most of the "unique" adjectives to describe the non-redundant mutations.

1. Page 6. Unclear "All but one clusters contained cells co‐migrated with cells of their sources.""Among 113 mutation isoforms, the major histocompatibility complex (HLA) was the most prominent with 68 iterations (60.2%) (Supplemental Table 3, Figure 3B)" There is nothing about HLA in Figure 3B.

Answer: We revised the sentence as "Cells in all but one clusters co-migrated with cells of their sources". The mutation isoform expressions were listed in supplemental Table 3. They are too small and become unreadable when put in the figure.

1. Page 10 "genes or isoforms that across all samples had with expression standard deviations less than" - probably "with" should not be there.

Answer: We correct the error and thank the reviewer for the comment.

1. Page 11 "UMAP analysis was performed using genes with standard deviations {greater than or equal to} 1.0 (182 wild‐type genes) and standard deviations >0.4 (282 mutated genes)". What do "wild-type" and "mutated" mean here?

Answer: We edited as "UMAP analysis was performed using gene expressions with standard deviations ≥ 1.0 (182 non-mutated genes) and gene mutation expression with standard deviations 0.4 (282 mutated genes)."

1. I could not find the description of Supplementary Tables.

Answer: The supplemental table legends are added in the revised manuscript.

1. In the Discussion section, the authors mention that mutations were mainly expressed in a specific isoform of a gene for a given cell. I suggest to emphasize this point in the Results section and illustrate it with a comparison of abundance of mutated and non-mutated isoforms

Answer: For HLA molecules, their expression appeared to be restricted to one known isoform, regardless of mutation status. This sentence is removed in the revision. A new section of DOCK8 and STEAP4 mutation expression is added to the result.

1. It is also mentioned that mutations may have an impact on the RNA splicing process. The authors should compare the observed isoform ratio to a prediction of the effect of variants on splicing by SpliceAI or similar tools

Answer: This sentence was removed from the discussion.

1. Figure 3c: triangles corresponding to HLA-positive cells are hard to distinguish

Answer: We provide a larger representation of the triangle and circle in figure 3c in the revision.

**Reviewer #2 (Recommendations For The Authors):**
Many of my comments could be addressed by spending time to provide the code/data and a walkthrough of analyses so that other users would be able to answer these questions on their own.

Answer: We have included a script section in the revision to ensure the reproducibility of the analysis. The raw data had been uploaded to GEO (see Methods).